# Algorithm Design for Edge Detection of High-Speed Moving Target Image under Noisy Environment

**DOI:** 10.3390/s19020343

**Published:** 2019-01-16

**Authors:** Fangfang Han, Bin Liu, Junchao Zhu, Baofeng Zhang

**Affiliations:** Tianjin Key Laboratory for Control Theory & Applications in Complicated Systems, School of Electrical and Electronic Engineering, Tianjin University of Technology, Tianjin 300384, China; zhujunchao_tjut@163.com (J.Z.); zhangbaofeng@tjut.edu.cn (B.Z.)

**Keywords:** high-speed vision, edge detection, noise suppression, wavelet filter, activation function

## Abstract

For some measurement and detection applications based on video (sequence images), if the exposure time of camera is not suitable with the motion speed of the photographed target, fuzzy edges will be produced in the image, and some poor lighting condition will aggravate this edge blur phenomena. Especially, the existence of noise in industrial field environment makes the extraction of fuzzy edges become a more difficult problem when analyzing the posture of a high-speed moving target. Because noise and edge are always both the kind of high-frequency information, it is difficult to make trade-offs only by frequency bands. In this paper, a noise-tolerant edge detection method based on the correlation relationship between layers of wavelet transform coefficients is proposed. The goal of the paper is not to recover a clean image from a noisy observation, but to make a trade-off judgment for noise and edge signal directly according to the characteristics of wavelet transform coefficients, to realize the extraction of edge information from a noisy image directly. According to the wavelet coefficients tree and the Lipschitz exponent property of noise, the idea of neural network activation function is adopted to design the activation judgment method of wavelet coefficients. Then the significant wavelet coefficients can be retained. At the same time, the non-significant coefficients were removed according to the method of judgment of isolated coefficients. On the other hand, based on the design of Daubechies orthogonal compactly-supported wavelet filter, rational coefficients wavelet filters can be designed by increasing free variables. By reducing the vanishing moments of wavelet filters, more high-frequency information can be retained in the wavelet transform fields, which is benefit to the application of edge detection. For a noisy image of high-speed moving targets with fuzzy edges, by using the length 8-4 rational coefficients biorthogonal wavelet filters and the algorithm proposed in this paper, edge information could be detected clearly. Results of multiple groups of comparative experiments have shown that the edge detection effect of the proposed algorithm in this paper has the obvious superiority.

## 1. Introduction

In recent years, the concept of video-metrics has been proposed [1]. Its processing object is mainly digital video (sequence images), and studies the measurement and estimation of the structural or motion parameters of the moving target [2,3]. A high-speed visual measurement system captures the status change of the moving target at the transient time by using a high-speed camera, and instantaneously presents the information that cannot be captured by human eyes or other conventional sensors, in the form of continuous sequence images. Therefore, the high-speed visual measurement system has important research value in the dynamic posture and parameter measurement of high-speed moving target, research of collision transient, weak signal extraction, and vibration analysis [4,5,6].

However, in order to capture the high-speed moving target image, the camera is required to have a fast-enough exposure frequency. When the camera’s exposure time is not shot enough and the moving target has a certain displacement during the exposure time, the image of moving target will be blurred. On the other hand, when the camera exposure time is too short, it needs to have enough light conditions to ensure the image quality, otherwise the image will also be blurred due to insufficient light. In addition, when using the fixed-focus lens for imaging, the movement of the high-speed moving target at the instant of capturing may be separated from the focal plane, resulting in defocusing blurring. Therefore, the above reasons often lead to the edge ambiguity of high-speed moving target image in high-speed camera measurement, which brings some difficulties for the accurate extraction of image features. Especially for the noisy environment of industrial applications such as high-speed moving target detection and measurement, noise signal makes the extraction of fuzzy image edge becomes a more difficult problem.

It should be noted that, for any optical imaging system, the aperture of the imaging system would always introduce diffraction effect and causing edge to be natural blurred, and the method proposed in this paper is based on assumption for a simplifying mathematical model. That is, for one clearly digital image, gray value of one image edge pixel will present in the form of a ‘step function’ relative to the other adjacent background pixels. While in some cases, for one blur digital image, gray value of one image edge pixel will be approximated in the form of a ‘slope line function’ relative to the other adjacent background pixels.

Edge is an important feature of image, so edge detection is an important algorithm in image analysis. For edge detection of digital image, a difficult problem is the edge detection in a noisy environment. Image denoising is a classical yet still active topic since it is an indispensable step in many practical applications. The goal of image denoising is to recover a clean image from a noisy observation. From a Bayesian viewpoint, when the likelihood is known, the image prior modeling will play a central role in image denoising. Over the past few decades, various models have been exploited for modeling image priors, including nonlocal self-similarity (NSS) models [7,8,9,10,11], sparse models [12,13,14], gradient models [15,16,17] and Markov random field (MRF) models [18,19,20]. In particular, the NSS models are popular in state-of-the-art methods such as BM3D [8], LSSC [10], NCSR [13] and WNNM [21]. However, despite their high denoising quality, most of the denoising methods typically suffer from two major drawbacks. First, those methods generally involve a complex optimization problem in the testing stage, making the denoising process time-consuming [13,22]. Thus, most of the methods can hardly achieve high performance without sacrificing computational efficiency. Second, the models in general are non-convex and involve several manually chosen parameters, providing some leeway to boost denoising performance. To overcome the above drawbacks, several discriminative learning methods have been recently developed to learn image prior models in the context of truncated inference procedure. The resulting models are able to get rid of the iterative optimization procedure in the test phase. Reference [23] constructed a feed-forward denoising convolutional neural networks (DnCNNs), which utilized residual learning and batch normalization to speed up the training process as well as boost the denoising performance. The DnCNN model can not only exhibit high effectiveness in several general image denoising tasks such as Gaussian denoising, single image super-resolution, and JPEG image deblocking, but also be efficiently implemented by benefiting from GPU computing. Reference [24] trained a set of fast and effective CNN (convolutional neural network) denoisers, and integrated them into model-based optimization method to solve some image restoration problems. Reference [25] proposed a novel model which uses a devised cost function involving semisupervised learning based on a large amount of corrupted image data with a few labeled training samples, to effectively eliminate the visual effects generated by the impulse noise from the corrupted images. 

However, the goal of the paper is not to recover a clean image from a noisy observation, but to make a trade-off judgment for noise signal and image edge information directly according to the characteristics of wavelet transform coefficients, to realize the extraction of edge information from a noisy image directly. Because image noise and image edge are both high-frequency information, it is difficult to make a choice with frequency band only. Here, ‘high-frequency information’ means spatial frequency information; that is, for one digital image, edge information and noise information both belong to the mutation region of pixel gray value in spatial distribution. This paper only discusses edge detection techniques for one single frame image, so it has no relation with time frequency. The aim of edge detection is to accurately extract edge information with a better noise suppression effect. 

From a mathematical point of view, the image edge presents local singularity (a function is discontinuous at some point or its derivative is discontinuous, and the function is singular at this point), and the singularity index is measured by the Lipschitz index [26]. Up to now, Fourier transform is still the main mathematical tool for the analysis of singularity, but the global idea of Fourier transform is not suitable for local singularity detection. In comparison, the wavelet transform with ‘zoom’ function is very suitable for signal singularity detection. Furthermore, there is a certain interlayer correlation relationship between wavelet transform coefficients. The wavelet coefficients of different scales form a kind of relationship as a wavelet tree. The number of the singular points corresponding to the texture information increases with the increase of the scale, and has the positive Lipschizt index property; while the number of points corresponding to noise signal decreases with the increase of the scale, and has negative Lipschizt index property [26,27,28,29]. This provides a strong theoretical foundation for edge detection in noise environment based on wavelet transform.

On the other hand, the application effect of wavelet transform is dependent on the choice of wavelet function [30,31]. It can be seen from the subsequent experiments in this paper that different wavelet functions used by the same algorithm will make different effects. However, the biggest difference between the wavelet function with other functions discussed in the past is that most of the wavelet functions are difficult to find the obvious expression. In practical applications, the properties of the wavelet function are often determined only by the properties of the graph or the values of the scale function at several discrete points. Therefore, it is necessary to study wavelet filter design which meets certain characteristics. In addition, most of the wavelet filter coefficients that satisfy certain characteristics are irrational coefficients, but the irrational coefficient brings a lot of inconvenience to the computer application, especially the truncation error causing precision loss. Therefore, it has certain significance to study the coefficients wavelet filter design.

The aim of this paper is to realize image edge detection in noise environment directly. It does not take into account the low-frequency information of the image overview, only to make a judgment for the coefficients in the high-frequency fields of wavelet transform. This paper proposes a method to use the correlation relationship between wavelet transform coefficients of different scales to detect edge information with suppression of noise information. First, from bottom to top, the important coefficients are preserved by using the activation function of neural network unit; secondly, from top to bottom, the unimportant coefficients are eliminated by the outlier search judgment algorithm. In addition, based on the design theory of Daubechies orthogonal compactly-supported wavelet filters, rational coefficients wavelet filter can be designed by increasing free variables. Experiments show that the fuzzy image edge detection of high-speed moving targets under noise environment can be solved by using the length 8-4 rational coefficients biorthogonal wavelet filter and the edge detection algorithm proposed in this paper.

## 2. Principle of Noise-Tolerant Edge Detection Algorithm

### 2.1. Theoretical Basis

**Definition** **1.**
*If a two-dimensional function θ(x,y) satisfies $\int^{\text{​}}\int_{R^{2}}\theta \left(x,y\right)dxdy\neq 0$, $\underset{x,y\to \pm \infty }{lim}\theta \left(x,y\right)=0$, and θ(x,y)≥0, then it is called a two-dimensional smooth function.*

*Make $\theta_{s}\left(x,y\right)=\frac{1}{s}\theta \left[\frac{x}{s},\frac{y}{s}\right]$ is the function on the scale s of θ(x,y), the smoothness of two-dimensional signals f(x,y) is achieved by convolution on different scales s with $\theta_{s}\left(x,y\right)$, which is expressed as*
(1)$$\left(f*\theta_{s}\right)\left(x,y\right)=\int_{R}\int_{R}f\left(x-u,y-v\right)\theta_{s}\left(u,v\right)dudv$$
$\left(f*\theta_{s}\right)\left(x,y\right)$
*, which is the convolution of*
f(x,y)
*and $\theta_{s}\left(x,y\right)$, attenuates the high frequency signal of f(x,y) without changing the low frequency signal, thus smoothing the function of f(x,y). Therefore, the edge of a square integrable function $f\left(x,y\right)\in L^{2}\left(R\right)$ under the scale s is defined as the local mutation point after f(x,y) is smoothed by $\theta_{s}\left(x,y\right)$.*


**Definition** **2.**
*Let Wf(s,t) be denoted as the continuous wavelet transform of f(t) under the scale s. On the scale of $s_{0}$.*
(1)
*$\frac{\partial Wf\left(s_{0},t\right)}{\partial t}$ is the derivative of $Wf\left(s_{0},t\right)$ with respect to t, if it is equal to zero at $t=t_{0}$, then the wavelet transform $Wf\left(s_{0},t\right)$ has local extreme value in $\left(s_{0},t_{0}\right)$.*
(2)
*If for any point t in the neighborhood of point $t_{0}$ satisfies $\left|Wf\left(s_{0},t\right)\right|\leq \left|Wf\left(s_{0},t_{0}\right)\right|$, and in the left neighborhood or the right neighborhood strictly meet the above inequality relations, $\left(s_{0},t_{0}\right)$ is called the maximum point of wavelet transform modulus |Wf(s,t)| in the scale of $s_{0}$; $\left|Wf\left(s_{0},t_{0}\right)\right|$ is called the maximum modulus of wavelet transform modulus |Wf(s,t)| on the point of $\left(s_{0},t_{0}\right)$. *
(3)
*On the plane of $\left(s_{0},t\right)$, if there is a curve that every point is a modulus maximum point of $\left|Wf\left(s_{0},t\right)\right|$, it is called a modulus maximum curve.*


*Assuming that T is a threshold value (T>0), the point $t_{0}$ that meets the following two conditions under the scale $s_{0}>0$ is called the edge point of the signal under the scale $s_{0}$:*
(1)
$\left|Wf\left(s_{0},t_{0}\right)\right|\geq T$
(2)
*$\left|Wf\left(s_{0},t\right)\right|$ gets the local modulus maximum at point $t_{0}$.*



**Definition** **3.**
*Assuming that $f\left(x\right)\in L^{2}\left(R\right)$, then say the function f(x) has the Lipschitz exponent at $x_{0}$, it means that there is a constant K for $\forall x\in Bx_{0}$ ($Bx_{0}$ is an arbitrary open neighborhood of $x_{0}$) to make $\left|f\left(x\right)-f\left(x_{0}\right)\right|\leq K{\left|x-x_{0}\right|}^{\alpha }$.*

*The bigger α is, the smoother the function is, and the less singular it is; on the contrary, the smaller α is, the more dramatic the change of the function at that point, namely, the greater the singularity.*


Studies have shown that Lipschitz exponent with different properties for signal and noise [32]. In different scales of wavelet transform, the number of singular points with maximum modulus corresponding to the texture information increases with the increase of scale and has positive Lipschitz exponent property; but the points with maximum modulus corresponding to the standard Gaussian white noise become more sparse with the increase of scale, and has negative Lipschitz exponent property. Moreover, the wavelet transform direction of adjacent edge points is similar, while the wavelet transform direction of noise is random. All of these provide a favorable basis for edge detection based on wavelet transform.

### 2.2. Algorithm Basis

In this paper, an edge detection algorithm based on in-layer coefficients characteristics and interlayer coefficients correlation of wavelet transform is proposed, which is mainly based on the following theoretical basis:(1)Singular point of signal usually corresponds to the maximum value of the wavelet transform coefficient.(2)There is a correlation relationship between wavelet coefficients in different scales, that is, when the parent coefficient has a larger value, its four sub-coefficients also have a larger value.(3)Noise and edge signal have a different Lipschitz exponent property, which means that wavelet transform modulus of noise decreases with the increase of scale, and wavelet transform modulus of image edge signal increases with the increase of scale.

The general process of noise-tolerant edge detection algorithm proposed in this paper is shown in Figure 1. The specific steps show as follows.(1)First, select a kind of wavelet filter. The application effect of wavelet transform is closely dependent on the selection of wavelet function. Wavelet selection should comprehensively consider various performances according to the application requirements. Generally speaking, at a certain scale, the number of the detected extreme points has a linear relationship with the number of wavelet disappearing moments. The aim of this paper is to do edge detection, and a filter with small vanishing moments can be selected to get more high-frequency information in transformation domain. See Section 3 of this paper for detailed discussion.(2)Second, wavelet transform is performed at a certain scale. For digital images, the number of wavelet coefficients decreases by ration of 2×2. If the original image is small, the coefficients after several layers of wavelet transformation will become too less to support the detection and analysis algorithm. In this paper, for image with resolution of 640×640, three-layer wavelet transform is adopted for coefficients analysis.(3)The correlation relationship between wavelet transform coefficients is used for edge detection and denoising. This is composed of two steps: the first, from bottom to top, preserve the important coefficients by using the idea of activation function of neural network unit; the second, from top to bottom, eliminate unimportant coefficients by using outlier point search judgment algorithm. See Section 2.3 of this article for detailed algorithm.(4)Wavelet inverse transform is applied to the coefficients preserved after the above treatment.(5)According to the actual processing effect, some mathematical morphology algorithm can be adopted to remove the isolated noise points, so as to obtain a clearer and complete edge image.

### 2.3. Process of Denoising by Interlayer Correlation of Wavelet Coefficients

An important feature of wavelet transform coefficients is that the coefficients of each layer do not exist in isolation, but have a certain interlayer correlation. With the increase of wavelet transform scale, the number of wavelet coefficients is becoming less, but those coefficients contain more information, so they have more significance and their importance is greater. Furthermore, wavelet coefficients of different scales form a relationship as a wavelet tree [26]. That is, the wavelet coefficients of a certain scale are correlated with the four leaf coefficients of the next scale. According to wavelet tree theory, if a certain coefficient has a larger value, its four leaf coefficients also have larger values. As shown in Figure 2, taking three-layer wavelet transform as an example, if a coefficient of the third-layer transform’s space of $HL_{2}$ has a larger value, then the four-leaf coefficients of the second-layer transform’s space of $HL_{1}$ will have larger values, and the 16 leaf coefficients of the first-layer transform’s space of $HL_{0}$ in corresponding position will also have larger values. On the other hand, according to the Lipschitz exponent property of signal and noise, the modulus of image edge signal increases with the increase of scale, while the modulus of noise decreases with the increase of scale. 

In Figure 2, L means for ‘low-pass’, H means for ‘high-pass’; LH, HL, LL, and HH respectively mean ‘horizontal direction low-pass and vertical direction high-pass’, ‘horizontal direction high-pass and vertical direction low-pass’, ‘horizontal direction low-pass and vertical direction low-pass’ and ‘horizontal direction high-pass and vertical direction high-pass’ field of wavelet transform; the lower-index denotes different scales of wavelet transform.

#### 2.3.1. Process of Wavelet Coefficients Adopting Idea of Neural Network Activation Function

Activation function is a function running on the neurons of artificial neural network, which is responsible for mapping the input neurons to the output neurons end [33]. In one neuron, input elements are applied to the activation function after being weighted and summed to determine the output of neuron, as shown in Figure 3. The purpose of the activation function is to introduce the nonlinear features into the neural network [34,35,36]. 

The algorithm of activation function can be referred to several classic activation functions of neurons as shown in Figure 4: Sigmoid, ReLU, and Softplus [37,38].

Based on the idea of neuron activation function shown in Figure 3 and the interlayer correlation of wavelet coefficients shown in Figure 2, ‘activation operation’ (importance judgment) of wavelet coefficients proposed by this paper is shown in Figure 5. Start with the leaf coefficients of the lowest transformation layer, the weighted sum of the four leaf coefficients is used to determine the importance of the corresponding mother coefficient on the upper layer, and similarly, do the same operation at those high scale layers.

For the experiments in this article, the ‘step function’ is adopted as the activation function, that is, the results of the upper layer coefficients are ‘activated’ only by the binary state of taking and discarding. With further research, ReLU function can be considered to be adopted according to the experimental situation, for the higher layer coefficients can be utilized in the form of increasing weight.

Taking the symbols shown in Figure 5, the interlayer coefficients activation processing algorithm proposed in this paper is shown in Equation (2).

(2)$$f\left(x\right)=\{\begin{array}{c} 1\;x\geq thres\\ 0\;x<thres \end{array},\;\;thres=\sum^{\text{​}}w_{i,j}a_{i,j}+w_{i,j+1}a_{i,j+1}+w_{i+1,j}a_{i+1,j}+w_{i+1,j+1}a_{i+1,j+1}$$

In Equation (2), $a_{i,j}$ is one coefficient of one wavelet transform layer, $a_{i,j+1}$, $a_{i+1,j}$, $a_{i+1,j+1}$ are the neighbor coefficients of $a_{i,j}$, and $w_{i,j}$, $w_{i,j+1}$, $w_{i+1,j}$, $w_{i+1,j+1}$ are the weight values of the coefficient corresponding wavelet coefficients.

#### 2.3.2. Judgment for Isolated Coefficients

After the above important coefficients’ ‘activation’ treatment, coefficients binarization has been realized, and the important coefficients also have been reserved. In order to further eliminate the remaining noise points’ coefficients, this paper proposes a top-to-down interlayer search algorithm for isolated coefficients to eliminate the isolated noise. The process of the algorithm is shown in Figure 6.

Starting from one coefficient of a certain wavelet transform layer, if the coefficient’s value is 1, then make a judgment if it is an isolated coefficient, which means making a judgment whether its neighborhood coefficients are all 0. If it is an isolated coefficient, then its value is set to 0 (considering the wavelet coefficients corresponding to image edge usually do not appear in isolation), and at the same time, its four leaf coefficients corresponding to the next scale are all set to 0. While if it is not an isolated coefficient, make a judgment weather the value of its root coefficient on the upper scale is 0, if the root coefficient’s value is 0, it should also be set to 0, as well as its four leaf coefficients. Otherwise, when the above conditions are not met, the coefficient’s value should be reserved by 1. The overall workflow should be from large wavelet transform scale space to small wavelet transform scale space.

For isolated coefficients judgment, the eight neighborhood locations of an object coefficient are shown in Figure 7. In order to save search time, characteristics of different wavelet transform fields could be taken into account. For example, wavelet transform coefficients field of LH as shown in Figure 2, represents the image’s horizontal direction low-pass and vertical direction high-pass filtering information, and its coefficients mainly reflect the edge signal in the horizontal direction. So when judging whether a coefficient is an isolated coefficient, more attention could be paid to its neighborhood coefficients in the horizontal direction, that is, the coefficients at the locations of $l_{1},l_{8},l_{7},l_{3},l_{4},l_{5}$.Similarly, the field of $HL_{s}$ mainly reflects the edge information of the vertical direction, its coefficients’ six neighborhood locations of $l_{1},l_{2},l_{3},l_{5},l_{6},l_{7}$ should be pay more attention; the field of $HH_{s}$ mainly reflects the edge information of the diagonal direction, its coefficients’ four neighborhood locations of $l_{1},l_{3},l_{5},l_{7}$ should be pay more attention. Here the means of HL and HH refer to the Figure 2, and the lower-index s denotes a certain scale of wavelet transform.

Experimental results of this algorithm are shown in Section 4 of this paper.

## 3. Design of Rational Coefficients Biorthogonal Wavelet Filters

Wavelet filters that constructed based on Daubechies compactly-supported theory mostly have nonlinear phases and irrational coefficients. Filter with nonlinear phase is easy to cause distortion in image processing, while the irrational coefficients will bring much inconvenience to wavelet transform implemented on computer hardware. By the thought of complete reconstruction filter idea and adding vanishing moment characteristics, rational coefficients biorthogonal wavelet filters with linear phases can be designed.

### 3.1. Construction of Biorthogonal Complete Reconstruction Wavelet Filters

A biorthogonal complete reconstruction wavelet filter group [39,40] is composed of four filters, which are decomposition low-pass filter H(z), reconstruction low-pass filter $\tilde{H}\left(z\right)$, decomposition high-pass filter G(z) and reconstruction high-pass filter $\tilde{G}\left(z\right)$. To constitute a complete reconstruction filter group, the four filters should satisfy the following relationships, where $F_{0}\left(z\right)$ is called the deformation term and $F_{1}\left(z\right)$ is called the aliasing term.

(3)$$F_{0}\left(z\right)=H\left(z\right)\tilde{H}\left(z\right)+G\left(z\right)\tilde{G}\left(z\right)=2z^{-l}$$

(4)$$F_{1}\left(z\right)=H\left(-z\right)\tilde{H}\left(z\right)+G\left(-z\right)\tilde{G}\left(z\right)=0$$

According to the filter design principles shown as in Equations (3) and (4), this paper proposed a design scheme as

(5)$$\{\begin{array}{c} G\left(z\right)=z\tilde{H}\left(-z\right),\;\;that\;is,\;\;G\left(-z\right)=-z\tilde{H}\left(z\right)\\ \tilde{G}\left(z\right)=z^{-1}H\left(-z\right) \end{array}$$

In Equation (3), l represents time delay term. For one specified wavelet filter design, proposing setting the time delay term l=0, then Equations (3) and (4) can be represented as

(6)$$F_{0}\left(z\right)=H\left(z\right)\tilde{H}\left(z\right)+G\left(z\right)\tilde{G}\left(z\right)=2$$

(7)$$F_{1}\left(z\right)=H\left(-z\right)\tilde{H}\left(z\right)+G\left(-z\right)\tilde{G}\left(z\right)=0$$

Make
(8)$$Y\left(z\right)=H\left(z\right)\tilde{H}\left(z\right)$$

Then by derivation from Equations (5)–(8), Equation (9) can be obtained
(9)Y(z)+Y(−z)=2

Therefore, for the four filters, only two low-pass filters H(z) and $\tilde{H}\left(z\right)$ should be designed firstly, and two high-pass filters G(z) and $\tilde{G}\left(z\right)$ can be calculated according to Equation (5). Design of the symmetric and compactly-supported wavelet filters of H(z) and $\tilde{H}\left(z\right)$ will be discussed next.

### 3.2. Construction of Even Length Symmetric Compactly-Supported Biorthogonal Wavelet Filters

The original intention of construction of biorthogonal wavelet filters is to make wavelet filters not only have linear phase, but also constitute an orthogonal complete reconstruction system. A sufficient and necessary condition for filter’s linear phase is that the filter’s impulse response is symmetric about their center, that is $H\left(\omega \right)=e^{i2\lambda \omega }\bar{H}\left(\omega \right)$, or it is represented equivalently as the coefficients’ relationship as $h_{k}={\bar{h}}_{-2\lambda -k}$. When $h_{k}\in R$, coefficients can be viewed as taking −λ as the axis of symmetry [26]. 

Taking the design of an even length filter as an example, this paper proposes to make the wavelet low-pass filter to be symmetric about 1/2 axis or −1/2 axis, that is, the filter coefficients should satisfy $h_{k}=h_{1-k}$ or $h_{k}=h_{-1-k}$ [40,41]. Make decomposition filter symmetric about 1/2 axis and reconstruction filter symmetric about −1/2 axis respectively. Then two low-pass filters with even length symmetric biorthogonal structure can be designed as follows.

Suppose the decomposition low-pass filter (about 1/2 axis symmetry) and the reconstruction low-pass filter (about −1/2 axis symmetry) are represented by H(z) and $\tilde{H}\left(z\right)$ respectively, and their vanishing moments are represented by N and $\tilde{N}$ respectively.

Suppose
(10)$$T=z{\left(\frac{1+z^{-1}}{2}\right)}^{2}-\frac{1}{2}$$

Then
(11)$$H\left(T\right)=K_{1}\left(\frac{1+z}{2}\right){\left(T+\frac{1}{2}\right)}^{\frac{N-1}{2}}Q\left(T\right),\;\tilde{H}\left(T\right)=K_{2}\left(\frac{1+z^{-1}}{2}\right){\left(T+\frac{1}{2}\right)}^{\frac{\tilde{N}-1}{2}}\tilde{Q}\left(T\right)$$

Here, function Q is the higher order equation for the variable T, that is, function Q(T) can be represented as “$1+a_{1}T+a_{2}T^{2}+a_{3}T^{3}+\ldots +a_{m}T^{m}$”, and index m is determined by the wavelet filter vanishing moment. See the references for details of the theory of Daubechies orthogonal compactly-supported wavelet filter [26].

According to Equation (10), there is
$$\left(\frac{1+z}{2}\right)\left(\frac{1+z^{-1}}{2}\right)=T+\frac{1}{2}$$

Then
$$Y\left(T\right)=H\left(T\right)\tilde{H}\left(T\right)={\left(T+\frac{1}{2}\right)}^{\frac{N+\tilde{N}}{2}}Q\left(T\right)\tilde{Q}\left(T\right)$$
Y(T) needs to satisfy two conditions: ① complete reconstruction condition shown as Equation (9), ② Daubechies compactly-supported condition. 

For the complete reconstruction condition, combining Equations (9) and (10), and considering the symmetry of the filter, Y(T) needs to satisfy
(12)Y(T)+Y(−T)=1

For the Daubechies compactly-supported condition, suppose: $P\left(T\right)=Q\left(T\right)\tilde{Q}\left(T\right)$, and because $z=e^{i\omega }$, combines Euler’s formula and trigonometric function formula, P(T) is written with $x=\sin^{2}(\omega /2)$ in the form of
$$P\left(x\right)=Q\left(x\right)\tilde{Q}\left(x\right),\begin{array}{cc} & x=\sin^{2}\frac{\omega }{2} \end{array}$$

According to the solution of Taylor method, a set of special solutions can be obtained as
(13)$$P\left(x\right)=\overset{\frac{N+\tilde{N}}{2}-1}{\underset{n=0}{\sum }}C_{\frac{N+\tilde{N}}{2}-1+n}^{n}x^{n}+x^{\frac{N+\tilde{N}}{2}}r\left(x\right)$$

In the Equation (13), r(x) should satisfy
(14)r(x)+r(1−x)=0

P(x) should satisfy the Daubechies compactly-supported condition
(15)$$\underset{x\in \left[0,1\right],r\left(x\right)+r\left(1-x\right)=0}{max}\left|P\left(x\right)\right|<2^{2\cdot \left(\frac{N+\tilde{N}}{2}\right)-1}$$

### 3.3. Length 8-4 Rational Coefficients Symmetric Compactly-Supported Biorthogonal Wavelet Filters

Section 3.2 proposes the construction of orthogonal compactly-supported wavelets based on Daubechies method, and if the vanishing moment takes a larger value, that is Q(−1)≠0 and $\tilde{Q}\left(-1\right)\neq 0$, the wavelet function and the scale function established will have a fast attenuation rate, but wavelet filter coefficients obtained in this way are usually impossible to be rationalized. In order to obtain rational filter coefficients, one method is to reduce the number of vanishing moments to make some free variables appear in the equation, and by selecting appropriate values for those free variables, to make filter coefficients to be rationalized [26,39,40,41].

According to the vanishing moment constraint condition of biorthogonal wavelet filters, and in order to make free variables appear in the solution of Equation (12), select the vanishing moment $N=3,\tilde{\;N}=1$ respectively. According to calculation process given in Section 3.2, reasonable values of free variables can be obtained according to Daubechies compactly-supported condition expressed in Equation (15), and a rational coefficients biorthogonal wavelet filter group with length 8-4 can be obtained (see following reference [40] for filter design details).

The decomposition low-pass filter and reconstruction low-pass filters are
$$H\left(z\right)=-\left(1/96\right)z^{-3}-\left(5/96\right)z^{-2}+\left(21/96\right)z^{-1}+\left(81/96\right)+\left(81/96\right)z+\left(21/96\right)z^{2}-\left(5/96\right)z^{3}-\left(1/96\right)z^{4}$$
$$\tilde{H}\left(z\right)=-\left(1/4\right)z^{-2}+\left(5/4\right)z^{-1}+\left(5/4\right)-\left(1/4\right)z$$

The decomposition high-pass filter and reconstruction high-pass filters are
$$G\left(z\right)=-\left(1/4\right)z^{-1}-\left(5/4\right)+\left(5/4\right)z+\left(1/4\right)z^{2}$$
$$\tilde{G}\left(z\right)=\left(1/96\right)z^{-4}-\left(5/96\right)z^{-3}-\left(21/96\right)z^{-2}+\left(81/96\right)z^{-1}-\left(81/96\right)+\left(21/96\right)z+\left(5/96\right)z^{2}-\left(1/96\right)z^{3}$$

The decomposition scaling function, reconstruction scaling function, decomposition wavelet function, and reconstruction wavelet function derived from four filters are shown in Figure 8. The amplitude-frequency and phase-frequency characteristics of four filters are shown in Figure 9. It can be seen from Figure 9 that the four filters all have linear phases.

## 4. Experiments

### 4.1. Edge Detection for Images under Noisy Environments

A square image with uniform grey value in uniform value background is selected as the study object. The image without any noise is shown in Figure 10a. When Gaussian noise with mean value of 0 and variance of 0.01 is added to Figure 10a, the noisy image is shown in Figure 10b.

The operators of first order difference, Sobel, Laplacian, LoG, and Canny have been used for edge detection for Figure 10a and Figure 10b respectively. Edge detection results of Figure 10a are shown in Figure 11, and detection results of Figure 10b are shown in Figure 12.

According to the five result images in Figure 11, for the noiseless image, every operator could detect the image edge well. However, as can be seen from the five result images in Figure 12, all of the above operators are very sensitive to noise. As seen from Figure 12, Laplacian is the most sensitive operator to noise; LoG operator has been preprocessed by Gaussian smoothing, but a large number of noise points still exist; Canny operator has more edge search steps compared with other operators, the result will have some edge connection effects, so it is not suitable for noisy environment.

Using the edge detection algorithm proposed in Section 2.3 of this paper, does edge detection for image with Gaussian noise shown in Figure 10b, and the result is shown in Figure 13. In order to further discuss adaptabilities of this algorithm, edge detection experiments also have been carried out on the images with salt and pepper noise and speckle noise, as shown in Figure 14 and Figure 15, respectively.

Figure 13f, Figure 14f, and Figure 15f show the detection qualities by different color printing, white points are the final detected edge points, red rectangle is the enclosing rectangle of detected edge points and red small circle is the detected center of detected edge points, green rectangle is the original boundary of the original rectangle image and green small circle is the original center of original rectangle image.

Compared with each detection effect shown in Figure 12, it can be seen from Figure 13, Figure 14 and Figure 15 that the algorithm proposed in this paper could detect the image edge information well even in the noisy environment.

Table 1 shows the numerical results of above experiments. Table 1 gives the four vertex coordinates of the outer enclosing rectangle and the center point coordinates for Figure 13f, Figure 14f, and Figure 15f respectively, and the value of MSE gives the comprehensive mean square error of each index point in each case compared with each index point of the original image. Form Table 1, it can be seen that even in the noisy environment, the edge of the target could still be detected accurately and be located well.

### 4.2. Edge Detection for Noisy Image by Different Wavelet Filters

In the following experiments, there are six types of wavelet filters have been selected.

① Daubuchies D4 wavelet, an orthogonal wavelet, with nonlinear phase, a maximum vanishing moment and irrational wavelet filter coefficients. Waveforms of the scaling function and wavelet function are shown in Figure 16, and the wavelet filters are
$$H\left(z\right)=0.68291+1.1828z+0.31693z^{2}-0.18299z^{3}$$
$$G\left(z\right)=-0.18299z^{-2}-0.31693z^{-1}+1.1828-0.68291z$$

② CDF9-7 biorthogonal wavelets, with linear phases, maximum vanishing moments, and irrational wavelet filter coefficients. CDF9-7 wavelets have been recommended to be used in JPEG international image compression standards. Waveforms of the scaling functions and wavelet functions are shown in Figure 17, and the wavelet filters are
$$\begin{array}{ll} H\left(z\right) & =0.053498z^{-4}-0.033728z^{-3}-0.15645z^{-2}+0.53373z^{-1}+1.2059+0.53373z-0.15645z^{2}-0.033728z^{3}+0.053498z^{4}G\left(z\right)\\ & =0.091272z^{-2}-0.057544z^{-1}-0.59127+1.1151z-0.59127z^{2}-0.057544z^{3}+0.091272z^{4} \end{array}$$
$$\tilde{H}\left(z\right)=-0.091272z^{-3}-0.057544z^{-2}+0.59127z^{-1}+1.1151+0.59127z-0.057544z^{2}-0.091272z^{3}$$
$$\tilde{G}\left(z\right)=0.053498z^{-5}+0.033728z^{-4}-0.15645z^{-3}-0.53373z^{-2}+1.2059z^{-1}-0.53373-0.15645z+0.033728z^{2}+0.053498z^{3}$$

③ MATLAB Bior3.3 biorthogonal wavelets, with linear phases, maximum vanishing moments, and irrational wavelet filter coefficients. Waveforms of the scaling functions and wavelet functions are shown in Figure 18, and the wavelet filters are
$$H\left(z\right)=0.093735z^{-3}-0.2812z^{-2}-0.21872z^{-1}+1.406+1.406z-0.21872z^{2}-0.2812z^{3}+0.093735z^{4}$$
$$G\left(z\right)=0.24997z^{-1}-0.74989+0.74989z-0.24997z^{2}$$
$$\tilde{H}\left(z\right)=0.24997z^{-2}+0.74989z^{-1}+0.74989+0.24997z$$
$$\tilde{G}\left(z\right)=-0.093735z^{-4}-0.2812z^{-3}+0.21872z^{-2}+1.406z^{-1}-1.406-0.21872z+0.2812z^{2}+0.093735z^{3}$$

④ Length-4 rational coefficients orthogonal wavelet proposed by author in following reference [39], with nonlinear phase, but by decreasing of vanishing moment, the wavelet filters have rational coefficients. Waveforms of the scaling function and wavelet function are shown in Figure 19, and the wavelet filters are
$$H\left(z\right)=\left(12/17\right)+\left(20/17\right)z+\left(5/17\right)z^{2}-\left(3/17\right)z^{3}$$
$$G\left(z\right)=-\left(3/17\right)z^{-2}-\left(5/17\right)z^{-1}+\left(20/17\right)-\left(12/17\right)z$$

⑤ Length 7-5 rational coefficients biorthogonal wavelets proposed by author in following reference [41], with linear phases, and by decreasing of vanishing moments, the wavelet filters have rational coefficients. Waveforms of the scaling functions and wavelet functions are shown in Figure 20, and the wavelet filters are
$$H\left(z\right)=-\left(1/48\right)z^{-3}-\left(1/12\right)z^{-2}+\left(25/48\right)z^{-1}+\left(14/12\right)+\left(25/48\right)z-\left(1/12\right)z^{2}-\left(1/48\right)z^{3}$$
$$G\left(z\right)=-\left(1/8\right)z^{-1}-\left(1/2\right)+\left(5/4\right)z-\left(1/2\right)z^{2}-\left(1/8\right)z^{3}$$
$$\tilde{H}\left(z\right)=-\left(1/8\right)z^{-2}+\left(1/2\right)z^{-1}+\left(5/4\right)+\left(1/2\right)z-\left(1/8\right)z^{2}$$
$$\tilde{G}\left(z\right)=\left(1/48\right)z^{-4}-\left(1/12\right)z^{-3}-\left(25/48\right)z^{-2}+\left(14/12\right)z^{-1}-\left(25/48\right)-\left(1/12\right)z+\left(1/48\right)z^{2}$$

⑥ Length 8-4 rational coefficients biorthogonal wavelets proposed by author in following reference [41], with linear phases, and by decreasing of vanishing moments, the wavelet filters have rational coefficients. Waveforms of the scaling functions and wavelet functions are shown in Figure 8, and the wavelet filters are shown in Section 3.3 of this paper.

With the above six types of wavelet filters and the edge detection algorithm proposed in this paper, edge detection has been carried out for the image with Gaussian noise shown in Figure 10b, and the results are shown in Figure 21.

Seen from Figure 21, the results present obvious differences for different wavelets. Because of the nonlinear phases for wavelets, the reconstructed images for ① and ④ wavelets have a large distortion, and the edge information cannot be detected completely. For the same parameters of program, result of ③ wavelet shows the worst effect, because ③ wavelet retains the least information in high frequency region, while image edge detection is mainly based on high frequency information. In comparison, for linear phases, ②, ⑤, and ⑥ wavelets have better edge extraction effects. Among them, the result of ⑥ wavelet presents the best effect. Because vanishing moment of ⑥ wavelet has been reduced for the construction of the wavelet, so much high-frequency information could be remained for wavelet transform, which is conducive to image edge detection.

In addition, by comparing each image in Figure 21 and Figure 12, it can be seen that the edge detection method proposed in this paper based on the correlation relationship of wavelet coefficients can extract image edge information better in the noisy environment.

### 4.3. Edge Detection for Image with Gray-Gradient Edge under Noisy Environment

Above discussions are primarily for images with clear edge, but for the image edge with gradient gray value, how about the effects of edge detection algorithm proposed in this paper? Some experiments have been made in this paper. As shown in Figure 22, Figure 22a is a circle image with gray-gradient edge and a uniform gray background. Figure 22b is the gray value curve for pixels on the row of the middle position of the image. Figure 22c is the image added with Gaussian noise by the mean value of 0 and the variance of 0.01.

The above six types of wavelets have been used to perform edge detection for the gray-gradient circle image shown in Figure 22a and the noisy image shown in Figure 22c respectively. The edge detection algorithm adopts the method proposed in Section 2 of this paper, and the detection results are shown in Figure 23 and Figure 24 respectively. 

As can be seen from the detection results of each image in Figure 23, for the gray-gradient circle image without noise, all kinds of wavelets discussed in this paper can detect the gray-gradient edge well and show the edge’s width. However, as show in Figure 24, for the image with noise, no meaningful information can be detected by other wavelets except the ⑥ wavelet. Only by the length 8-4 rational coefficients biorthogonal wavelets and the algorithm proposed in this paper, gray-gradient edge in noisy image can still be detected well, and edge width could also be presented.

Figure 23 and Figure 24 also show the detection qualities by different color printing; white points are the final detected edge points; red rectangle is the enclosing rectangle of detected edge points and red small circle is the detected center of detected edge points; two green circles are the original outer and inner boundary of the original circle gray-gradient edge, and the green small circle is the original center of original circle image; blue circle is the fitting circle curve based on the detected edge points and fitting algorithm, and the small blue circle is the fitting circle center.

Table 2 shows the numerical results of Figure 23 and Figure 24. Table 2 gives the four vertex coordinates of the outer enclosing rectangle and the center point coordinates, fitting circle center coordinates and fitting circle radius for every result. The value of MSE gives the comprehensive mean square error of each index point in each case compared with each index point of the original image. Form Table 2 and Figure 24f, it can be seen that the wavelets of length 8-4 rational coefficients biorthogonal wavelets have obvious effect advantages, and even in the noisy environment, all the other wavelet filters are powerless, but by length 8-4 rational coefficients biorthogonal wavelets and edge detection algorithm proposed in this paper, gray-gradient edge information could still be detected accurately and be located well.

### 4.4. Edge Detection for High-Speed Moving Target Image under Noisy Environment

For experiments, some images for a high-speed moving Ping-Pong ball have been captured by the use of Japanese Photron AX200 high-speed camera, on a sunny afternoon and indoor lighting environment. Figure 25a is one frame image captured under the condition of frame rate with 3000 fps and resolution with 640 × 640; and Figure 25b is one frame image under the condition of frame rate with 5000fps and resolution with 640 × 640. Figure 25c,d are respectively the images of Figure 25a,b added with Gaussian noise by mean value of 0 and variance of 0.01.

The operators of first order difference, Sobel, Laplacian, LoG, Canny have been used for edge detection for every image in Figure 25, and detection results are shown in Figure 26, Figure 27, Figure 28 and Figure 29 respectively.

It can be seen from Figure 26 and Figure 27 that for the original captured images of high-speed moving targets, Sobel operator could generate a good edge detection effect, while Laplace operator, LoG operator, and Canny operator cannot detect the image edge well even under a relatively clean original image background. See from Figure 28 and Figure 29, image edge information is completely submerged in the noise signals and the operators are all powerless for edge detection.

For comparison, the above six types of wavelets have been used to perform edge detection for the four images of Figure 25. Results by the front five types are shown in Figure 30, Figure 31, Figure 32 and Figure 33, and results for the length 8-4 rational coefficients biorthogonal wavelets are shown in Figure 34, Figure 35, Figure 36 and Figure 37.

It can be seen from Figure 30, Figure 31, Figure 32 and Figure 33, for high-speed motion target, if the image has relatively clear quality, some wavelet filter could detect the edge information well; but if under a noisy environment, the five wavelet filter types above are unable to detect any meaningful information.

At last, adopt length 8-4 rational coefficients biorthogonal wavelets and the edge detection algorithm proposed in this paper, to do edge detection for the four images shown in Figure 25, and the results are presented in Figure 34, Figure 35, Figure 36 and Figure 37.

The four images of Figure 38 give the presentations of fitting circles and centers in their original images respectively. The fitting circles and centers are those calculated from edge detection results shown as in Figure 34d, Figure 35d, Figure 36d and Figure 37d.

It can be seen from the above experiments, for high-speed motion target image under noisy environment, several other wavelet filters are unable to detect any meaningful information, but the length 8-4 rational coefficients biorthogonal wavelet filters and with the edge detection algorithm proposed in this paper, could still detect the moving targets edge well and achieve a more accurate positioning of moving target.

## 5. Conclusions

In this paper, a noise-tolerant edge detection method based on the correlation relationship between layers of wavelet transform coefficients is proposed. According to the wavelet coefficients tree and the Lipschitz exponent property of noise, the neural network unit activation function was used to "activate" the relevant wavelet coefficients between layers, and then the important coefficients could be retained. In addition, an algorithm based on the judgment of isolated coefficients is designed to eliminate the isolated noise coefficients, so as to solve the hard problem of image edge detection under noisy environment. Experiments show that compared with other common edge detection operators, the proposed algorithm has obvious advantages.

On the other hand, the effects of different wavelet filters on edge detection are discussed. Based on the design theory of Daubechies orthogonal compactly-supported set wavelet filter, rational coefficients wavelet filters can be designed by increasing the number of free variables. By reducing the vanishing moments of wavelet filter, much high-frequency information can be retained in wavelet transform domain, which is suitable for edge detection. According to these two theories, with the designed length 8-4 rational coefficients symmetric compactly-supported biorthogonal wavelet filters and the edge detection algorithm proposed in this paper, some very prominent effects for image edge detection under noisy environment have been achieved, which have been demonstrated by multiple sets of contrast experiments.

On the basis of the theoretical research and simulation experiments, the actual high-speed moving target images acquired by high-speed camera have also been tested. Whether for the original acquisition images under the indoor environment, or for the images added strong noise, with the length 8-4 rational coefficients biorthogonal wavelet filters and the algorithm proposed in this paper, edge information of high-speed moving target image could be detected clearly, which is distinguished compared with other edge detection operators and other kinds of wavelet filters.

Therefore, the research content of this paper is of certain significance to the edge detection of high-speed moving target image, especially under noisy environment.

## Figures and Tables

**Figure 1 sensors-19-00343-f001:**
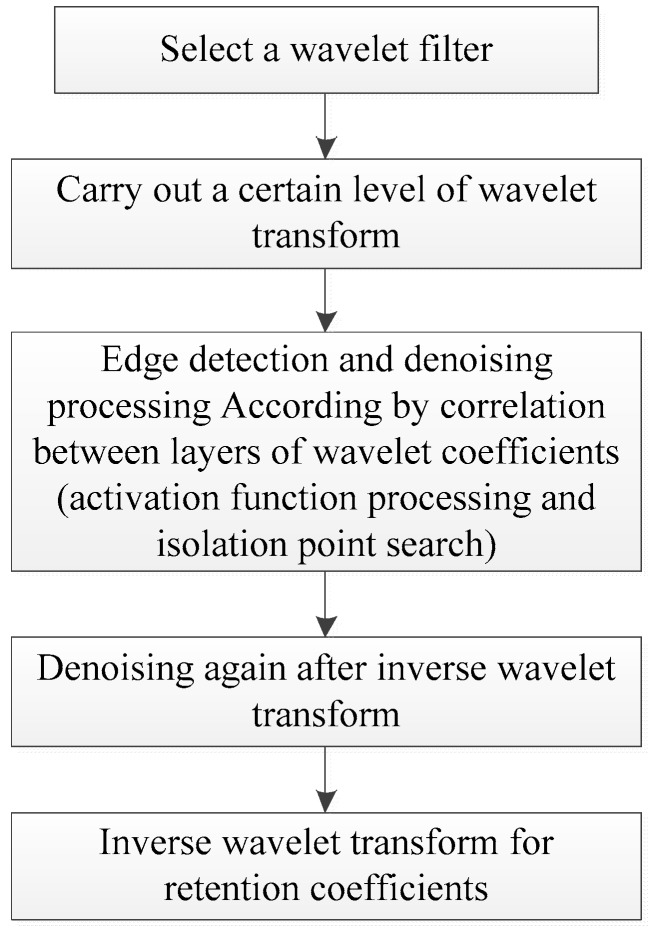
Noise-tolerant edge detection process based on wavelet transform.

**Figure 2 sensors-19-00343-f002:**
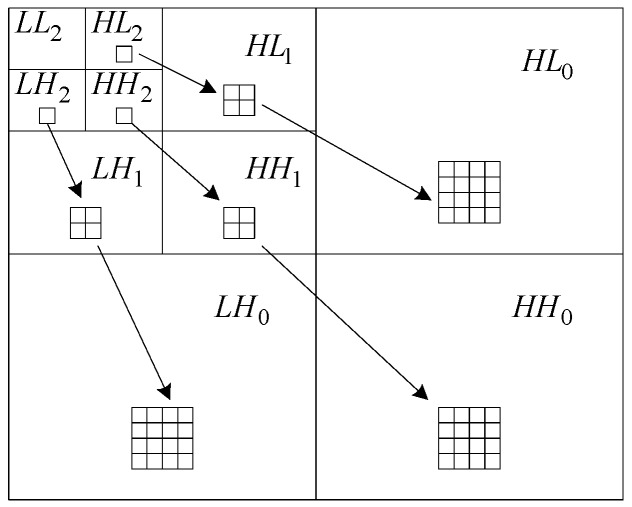
Correlation between layers of wavelet transform coefficients.

**Figure 3 sensors-19-00343-f003:**
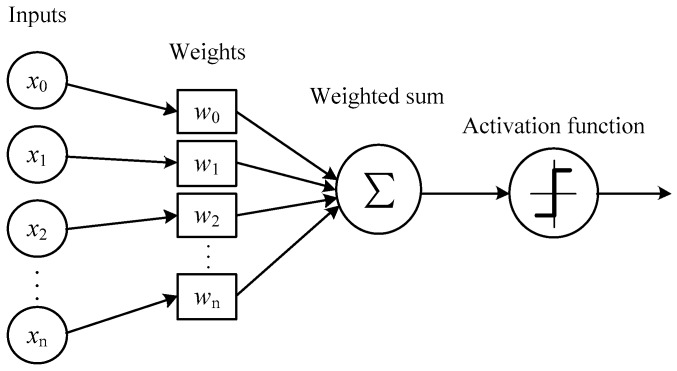
Schematic diagram of neuron activation function.

**Figure 4 sensors-19-00343-f004:**
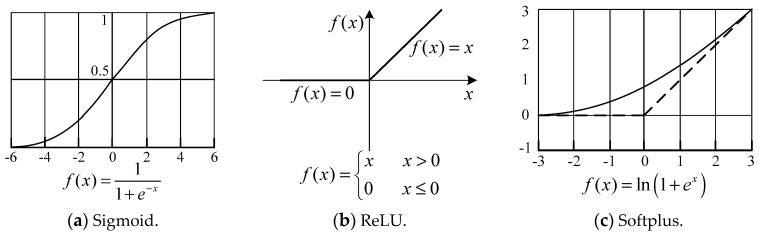
Curves of three types of neuronal activation functions.

**Figure 5 sensors-19-00343-f005:**
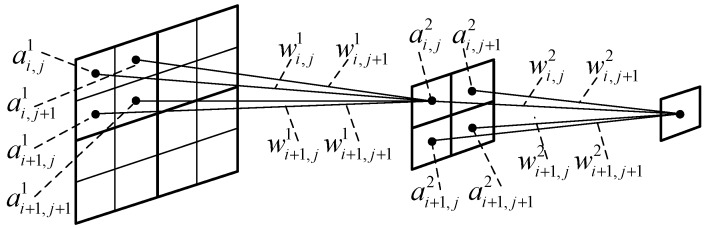
Schematic diagram of interlayer activation of wavelet coefficients.

**Figure 6 sensors-19-00343-f006:**
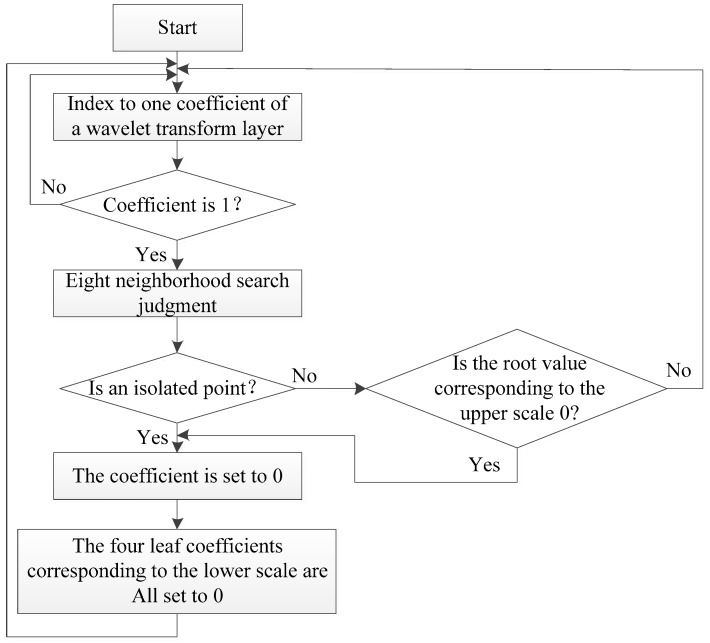
Noise deleting algorithm based on isolated coefficient judgment.

**Figure 7 sensors-19-00343-f007:**
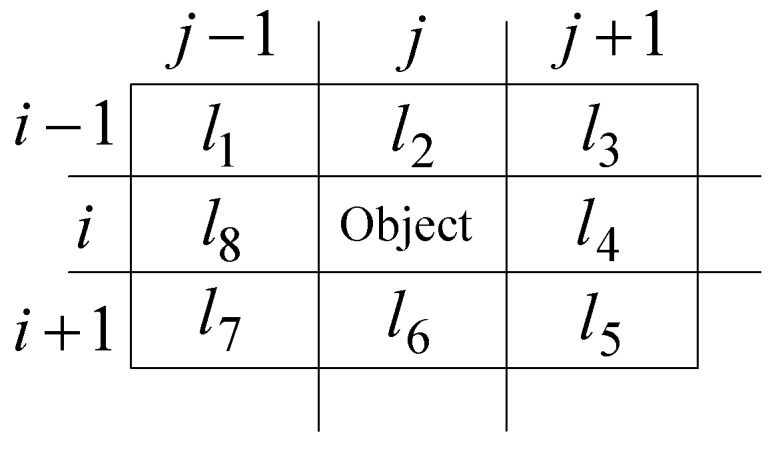
Eight neighborhoods and labels for target point.

**Figure 8 sensors-19-00343-f008:**
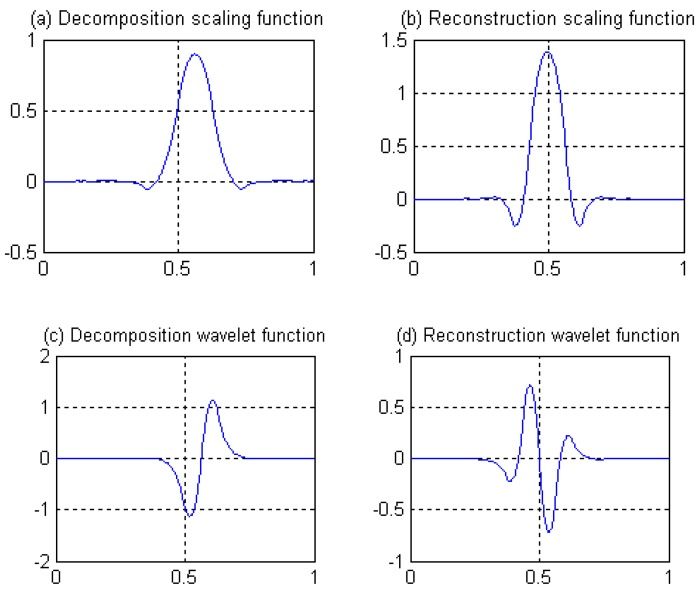
Scale function and wavelet function of length 8-4 rational coefficients biorthogonal wavelet filters.

**Figure 9 sensors-19-00343-f009:**
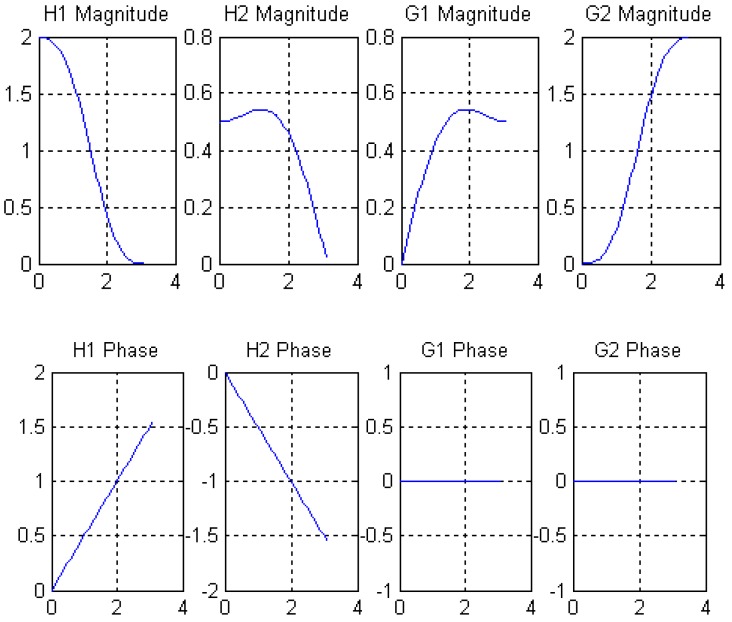
Amplitude-frequency and phase-frequency characteristics of length 8-4 rational coefficients biorthogonal wavelet filters.

**Figure 10 sensors-19-00343-f010:**
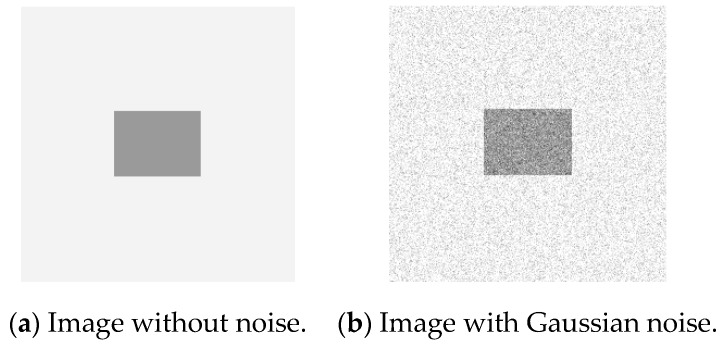
Original square images and its noisy image.

**Figure 11 sensors-19-00343-f011:**
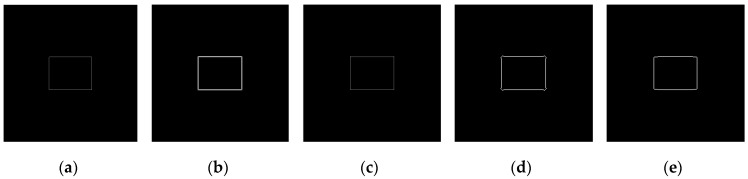
Edge detection results of noiseless images. (**a**) Detection results by first order difference operator. (**b**) Detection results by Sobel operator. (**c**) Detection results by Laplacian operator. (**d**) Detection results by LoG operator. (**e**) Detection results by Canny operator.

**Figure 12 sensors-19-00343-f012:**
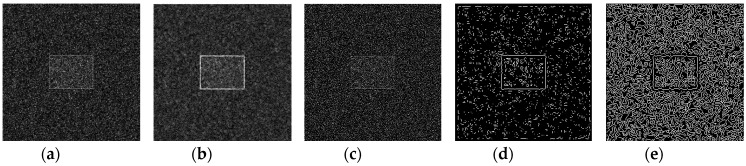
Edge detection results of noisy images. (**a**) Detection results by first order difference operator. (**b**) Detection results by Sobel operator. (**c**) Detection results by Laplacian operator. (**d**) Detection results by LoG operator. (**e**) Detection results by Canny operator.

**Figure 13 sensors-19-00343-f013:**
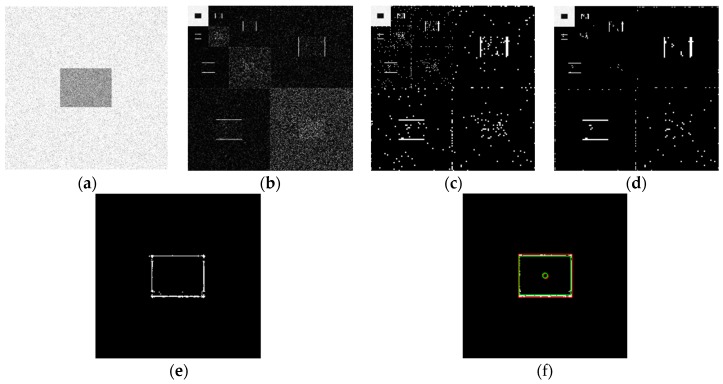
Edge detection for image with Gaussian noise by the algorithm proposed in this paper. (**a**) Original image with Gaussian noise. (**b**)Wavelet coefficients after three-layer transform. (**c**) Coefficients after activation and retention processing. (**d**) Coefficients after isolated judgment processing. (**e**) Final edge detection results. (**f**) Detection qualities by different color printings.

**Figure 14 sensors-19-00343-f014:**
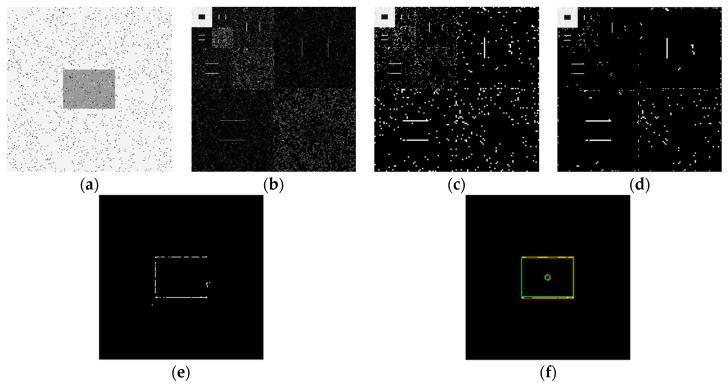
Edge detection for image with salt and pepper noise by the algorithm proposed in this paper. (**a**) Original image with salt and pepper noise. (**b**) Wavelet coefficients after three-layer transform. (**c**) Coefficients after activation and retention processing. (**d**) Coefficients after isolated judgment processing. (**e**) Final edge detection results. (**f**). Detection qualities by different color printings.

**Figure 15 sensors-19-00343-f015:**
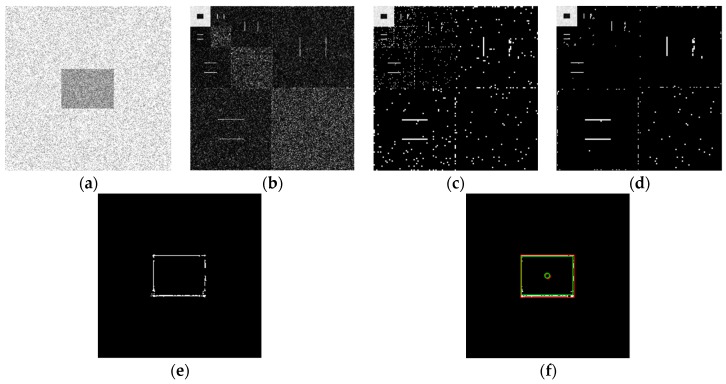
Edge detection for image with speckle noise by the algorithm proposed in this paper. (**a**) Original image with speckle noise. (**b**) Wavelet coefficients after three-layer transform. (**c**) Coefficients after activation and retention processing. (**d**) Coefficients after isolated judgment processing. (**e**) Final edge detection results. (**f**) Detection qualities by different color printings.

**Figure 16 sensors-19-00343-f016:**
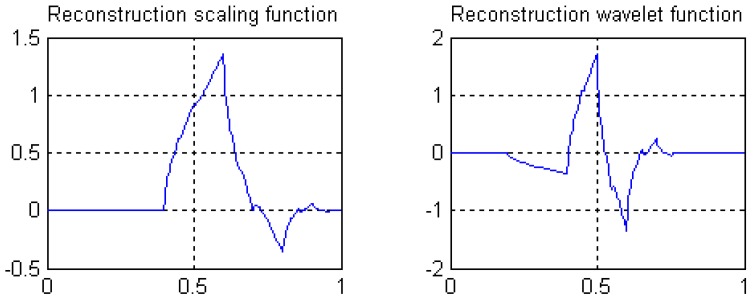
Waveforms of scaling function and wavelet function of Daubuchies D4 wavelet.

**Figure 17 sensors-19-00343-f017:**
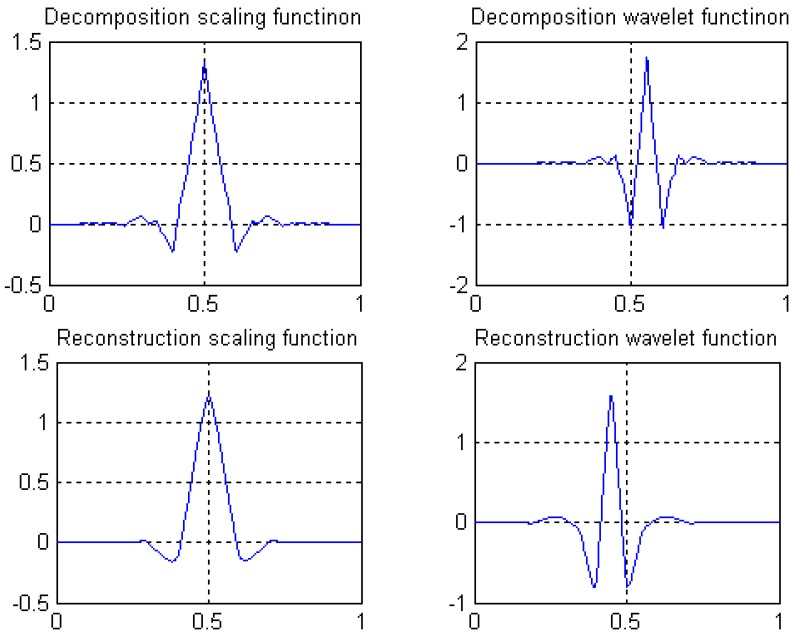
Waveforms of scaling functions and wavelet functions of CDF9-7 wavelets.

**Figure 18 sensors-19-00343-f018:**
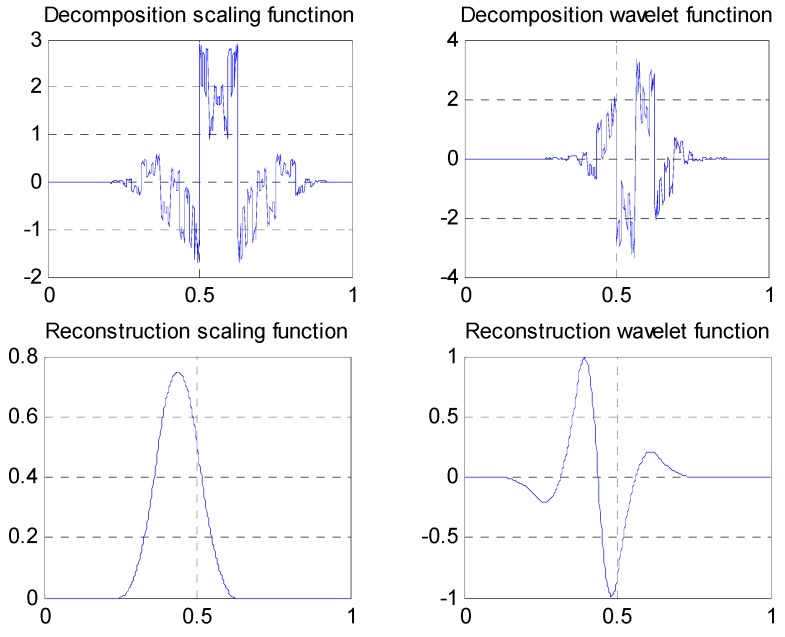
Waveforms of scaling functions and wavelet functions of MATLAB bior3.3 wavelets.

**Figure 19 sensors-19-00343-f019:**
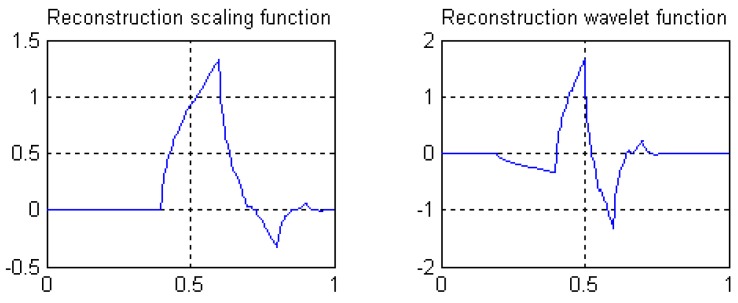
Waveforms of scaling function and wavelet function of length-4 rational-coefficients orthogonal wavelet.

**Figure 20 sensors-19-00343-f020:**
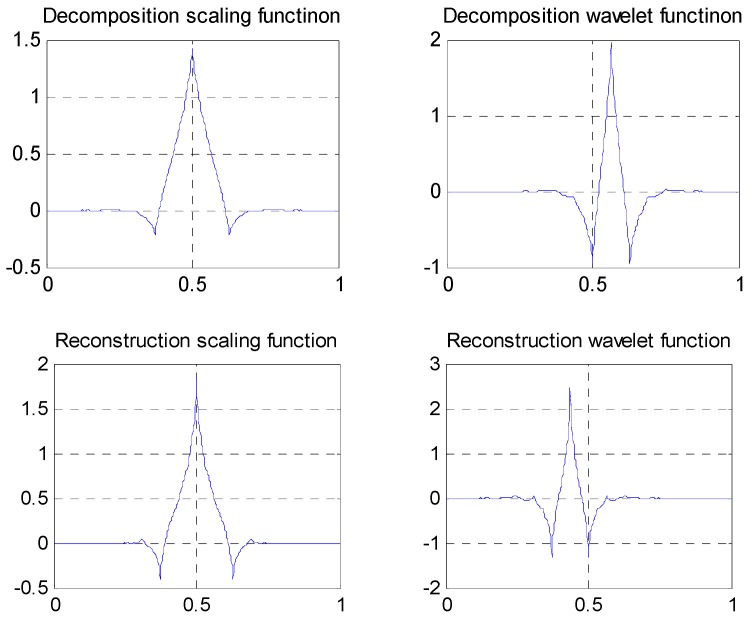
Waveforms of scale functions and wavelet functions of length 7-5 rational-coefficients biorthogonal wavelets.

**Figure 21 sensors-19-00343-f021:**
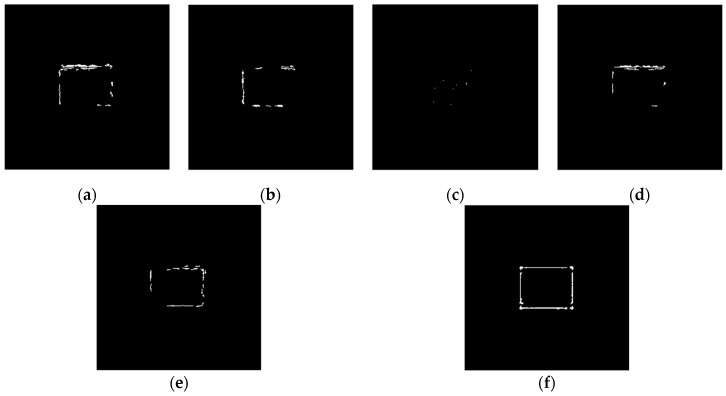
Results of edge detection by different wavelets for image with Gaussian noise. (**a**) Detection result by Daubuchies D4 wavelet. (**b**) Detection result by CDF9-7 biorthogonal wavelets. (**c**) Detection result by MATLAB Bior3.3 biorthogonal wavelets. (**d**) Detection result by length-4 rational coefficients orthogonal wavelet. (**e**) Detection result by length 7-5 rational coefficients biorthogonal wavelets. (**f**). Detection result by length 8-4 rational coefficients biorthogonal wavelets.

**Figure 22 sensors-19-00343-f022:**
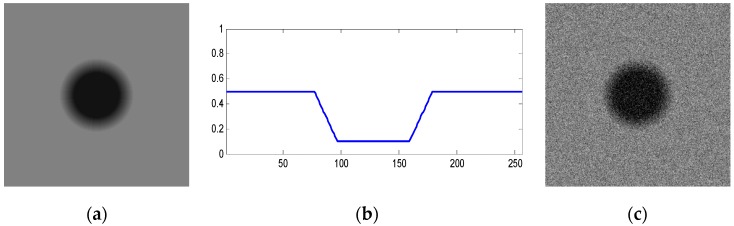
Gray gradient circle image and its noisy image. (**a**) Original gray-gradient edge. (**b**) Gray value curve of image middle line. (**c**) Gray-gradient edge image with Gaussian noise.

**Figure 23 sensors-19-00343-f023:**
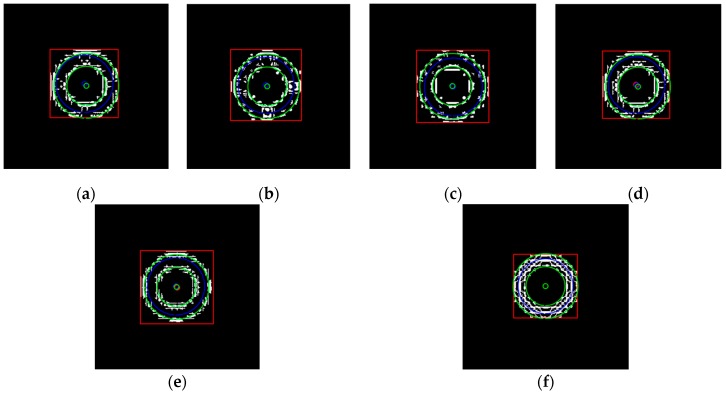
Edge detection of gray gradient edge circle image without noise. (**a**) Detection result by Daubuchies D4 wavelet. (**b**) Detection result by CDF9-7 biorthogonal wavelets. (**c**) Detection result by MATLAB Bior3.3 biorthogonal wavelets. (**d**) Detection result by length-4 rational coefficients orthogonal wavelet. (**e**) Detection result by length 7-5 rational coefficients biorthogonal wavelets. (**f**) Detection result by length 8-4 rational coefficients biorthogonal wavelets.

**Figure 24 sensors-19-00343-f024:**
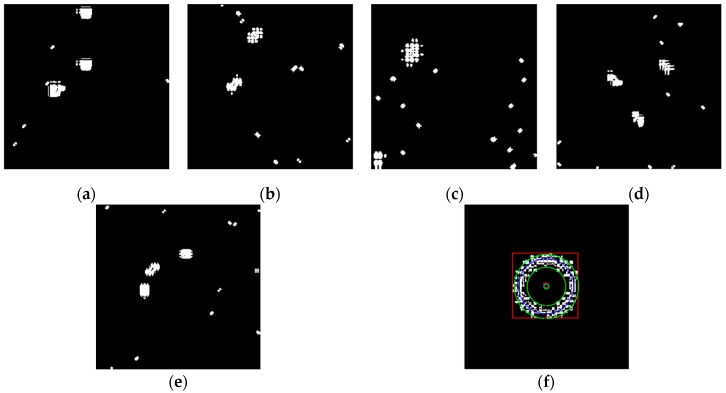
Edge detection of gray gradient edge circle image in noisy environment. (**a**) Detection result by Daubuchies D4 wavelet. (**b**) Detection result by CDF9-7 biorthogonal wavelets. (**c**) Detection result by MATLAB Bior3.3 biorthogonal wavelets. (**d**) Detection result by length-4 rational coefficients orthogonal wavelet. (**e**) Detection result by length 7-5 rational coefficients biorthogonal wavelets. (**f**) Detection result by length 8-4 rational coefficients biorthogonal wavelets.

**Figure 25 sensors-19-00343-f025:**
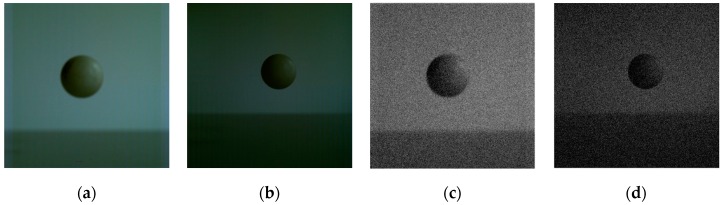
Original high-speed target images and their noisy images. (**a**) Captured with 3000 fps. (**b**) Captured with 5000 fps. (**c**) Image of 3000 fps with Gaussian noise. (**d**) Image of 5000 fps with Gaussian noise.

**Figure 26 sensors-19-00343-f026:**
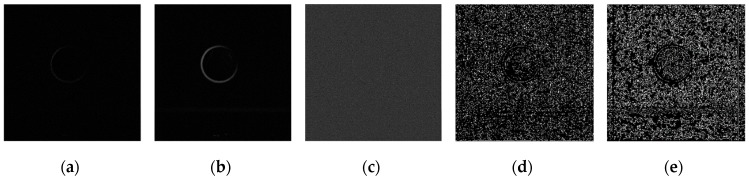
Detection results by several edge detection operators for Figure 25a. (**a**) Detection result by first order difference operator. (**b**) Detection result by Sobel operator. (**c**) Detection result by Laplacian operator. (**d**) Detection result by LoG operator. (**e**) Detection result by Canny operator.

**Figure 27 sensors-19-00343-f027:**
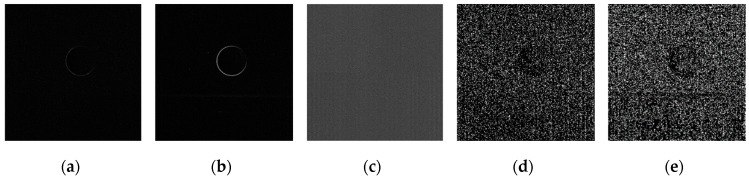
Detection results by several edge detection operators for Figure 25b. (**a**) Detection result by first order difference operator. (**b**) Detection result by Sobel operator. (**c**) Detection result by Laplacian operator. (**d**) Detection result by LoG operator. (**e**) Detection result by Canny operator.

**Figure 28 sensors-19-00343-f028:**
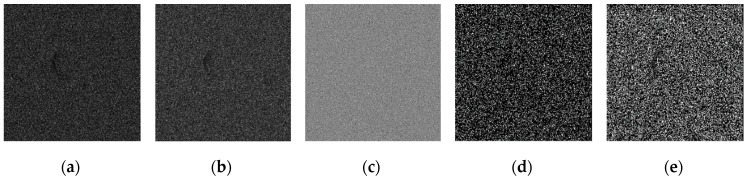
Detection results by several edge detection operators for Figure 25c. (**a**) Detection result by first order difference operator. (**b**) Detection result by Sobel operator. (**c**) Detection result by Laplacian operator. (**d**) Detection result by LoG operator. (**e**) Detection result by Canny operator.

**Figure 29 sensors-19-00343-f029:**
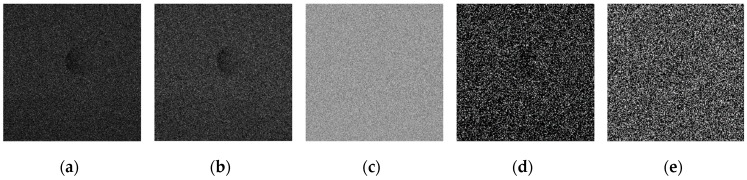
Detection results by several edge detection operators for Figure 25d. (**a**) Detection result by first order difference operator. (**b**) Detection result by Sobel operator. (**c**) Detection result by Laplacian operator. (**d**) Detection result by LoG operator. (**e**) Detection result by Canny operator.

**Figure 30 sensors-19-00343-f030:**
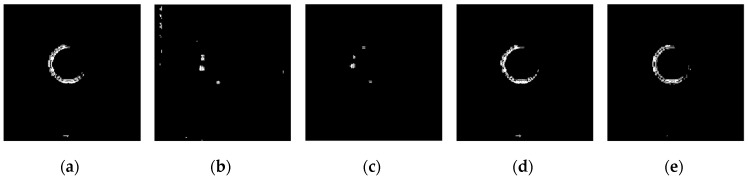
Detection results by five wavelet type filters for Figure 25a. (**a**) Detection result by Daubuchies D4 wavelet. (**b**) Detection result by CDF9-7 biorthogonal wavelets. (**c**) Detection result by MATLAB Bior3.3 biorthogonal wavelets. (**d**) Detection result by length-4 rational-coefficients orthogonal wavelets. (**e**) Detection result by length 7-5 rational coefficients biorthogonal wavelets.

**Figure 31 sensors-19-00343-f031:**
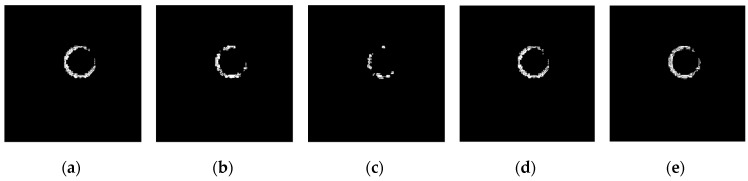
Detection results by five wavelet type filters for Figure 25b. (**a**) Detection result by Daubuchies D4 wavelet. (**b**) Detection result by CDF9-7 biorthogonal wavelets. (**c**) Detection result by MATLAB Bior3.3 biorthogonal wavelets. (**d**) Detection result by length-4 rational-coefficients orthogonal wavelets. (**e**) Detection result by length 7-5 rational coefficients biorthogonal wavelets.

**Figure 32 sensors-19-00343-f032:**
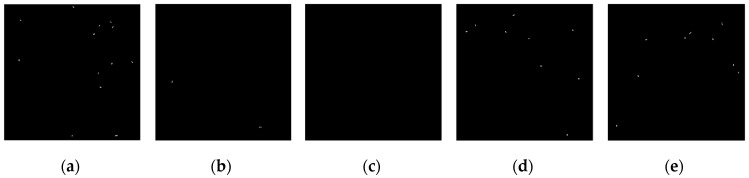
Detection results by five wavelet type filters for Figure 25c. (**a**) Detection result by Daubuchies D4 wavelet. (**b**) Detection result by CDF9-7 biorthogonal wavelets. (**c**) Detection result by MATLAB Bior3.3 biorthogonal wavelets. (**d**) Detection result by length-4 rational-coefficients orthogonal wavelets. (**e**) Detection result by length 7-5 rational coefficients biorthogonal wavelets.

**Figure 33 sensors-19-00343-f033:**
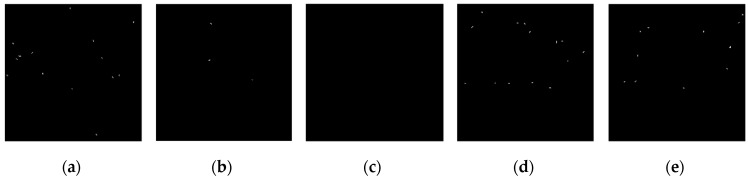
Detection results by five wavelet type filters for Figure 25d. (**a**) Detection result by Daubuchies D4 wavelet. (**b**) Detection result by CDF9-7 biorthogonal wavelets. (**c**) Detection result by MATLAB Bior3.3 biorthogonal wavelets. (**d**) Detection result by length-4 rational-coefficients orthogonal wavelets. (**e**) Detection result by length 7-5 rational coefficients biorthogonal wavelets.

**Figure 34 sensors-19-00343-f034:**
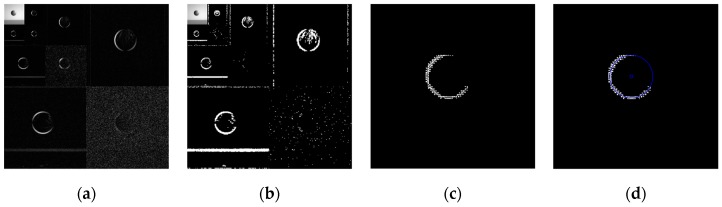
Detection result by length 8-4 rational coefficients biorthogonal wavelets for Figure 25a. (**a**) Wavelet coefficients after three-layer transform. (**b**) Wavelet coefficients after data processing. (**c**) Final edge detection result. (**d**) Fitting circle curve and center based on the edge detection result.

**Figure 35 sensors-19-00343-f035:**
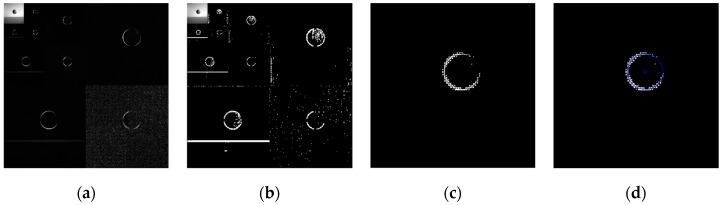
Detection result by length 8-4 rational coefficients biorthogonal wavelets for Figure 25b. (**a**) Wavelet coefficients after three-layer transform. (**b**) Wavelet coefficients after data processing. (**c**) Final edge detection result. (**d**) Fitting circle curve and center based on the edge detection result.

**Figure 36 sensors-19-00343-f036:**
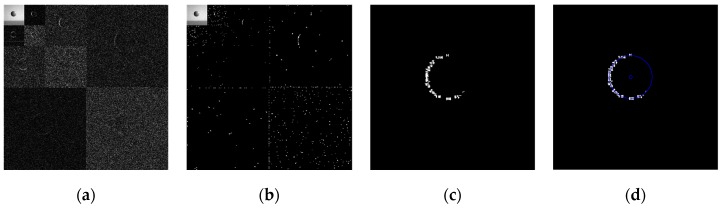
Detection result by length 8-4 rational coefficients biorthogonal wavelets for Figure 25c. (**a**) Wavelet coefficients after three-layer transform. (**b**) Wavelet coefficients after data processing. (**c**) Final edge detection result. (**d**) Fitting circle curve and center based on the edge detection result.

**Figure 37 sensors-19-00343-f037:**
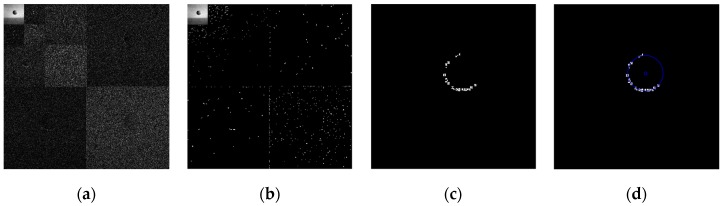
Detection result by length 8-4 rational coefficients biorthogonal wavelets for Figure 25d. (**a**) Wavelet coefficients after three-layer transform. (**b**) Wavelet coefficients after data processing. (**c**) Final edge detection result. (**d**) Fitting circle curve and center based on the edge detection result.

**Figure 38 sensors-19-00343-f038:**
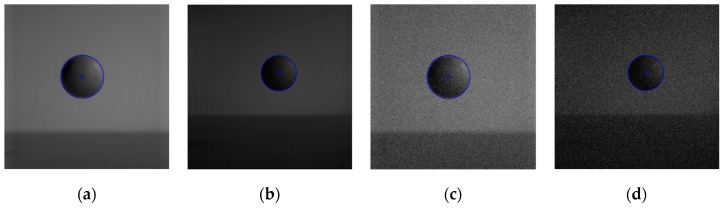
Locations of detection results on the original images. (**a**) Fitting circle and center presentation on Figure 25a. (**b**) Fitting circle and center presentation on Figure 25b. (**c**) Fitting circle and center presentation on Figure 25. (**d**) Fitting circle and center presentation on Figure 25d.

**Table 1 sensors-19-00343-t001:** Numerical results for rectangle edge detection with Gaussian, salt and pepper, and speckle noise.

	Left-Top	Right-Top	Right-Down	Left-Down	Center	MSE
**Original**	(98,88)	(98,168)	(158,168)	(158,88)	(128,128)	0
**Gaussian** **Figure 13f**	(95,87)	(95,170)	(162,170)	(162,87)	(128.5,128.5)	0.0460
**Salt and pepper** **Figure 15f**	(96,88)	(96,169)	(161,169)	(161,88)	(128.5,128.5)	0.0211
**Speckle** **Figure 15f**	(96,87)	(96,170)	(162,170	(162,87)	(129,128.5)	0.0424

**Table 2 sensors-19-00343-t002:** Numerical results for edge detection of gray-gradient edge circle image.

	Left-Top	Right-Top	Right-Down	Left-Down	Enclosing Rectangle Center	Fitting Circle Center	Fitting Circle Radius	MSE
**Original**	(78,78)	(78,178)	(178,178)	(178,78)	(128,128)	(128,128)	40	0
**Figure 23a**	(72,72)	(72,177)	(177,177)	(177,72)	(124.5,124.5)	(125.0,125.0)	43.88	0.2004
**Figure 23b**	(71,71)	(71,181)	(181,181)	(181,71)	(126,126)	(126.2,126.2)	43.58	0.2441
**Figure 23c**	(73,73)	(73,184)	(184,184)	(184,73)	(128.5,128.5)	(129.5,129.5)	44.48	0.1639
**Figure 23d**	(73,73)	(73,177)	(177,177)	(177,73)	(125,125)	(125.6,125.6)	43.64	0.1333
**Figure 23e**	(72,72)	(72,185)	(185,185)	(185,72)	(128.5,128.5)	(127.2,127.2)	44.32	0.2637
**Figure 23f**	(79,79)	(79,177)	(177,177)	(177,79)	(128,128)	(128.1,128.1)	42.20	0.0243
**Figure 24f**	(76,75)	(76,177)	(177,177)	(177,75)	(126.5,126)	(127.5,127.5)	42.97	0.0458

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
