# Peer review of "Algorithm Design for Edge Detection of High-Speed Moving Target Image under Noisy Environment"

_sensors, 2019, doi:10.3390/s19020343_

Round 1

Reviewer 1 Report

Manuscript ID: sensors-401640

Title: Algorithm Design for Edge Detection of High-speed Moving Target Image under Noisy Environment

Review Comments: Major revision

This paper deal with edge detection technique of high-speed moving target using neural network algorithm and wavelet transform. I would like to recommend the following contents for better organized paper.

It is good to compare with the existing edge detecting technique, but it is necessary to quantify and verify the superiority of the proposed edge detection technique.

It is necessary to describe the specific application fields of the proposed edge detection technique.

it is difficult to understand the relationship between equation 2-2 and figure 4 because of figure 5. So, equation 2-2 and figure 4 should be explained in one place and then figure 5 should be explain behind equation 2-2 and figure 4.

Author Response

Point 1: It is good to compare with the existing edge detecting technique, but it is necessary to quantify and verify the superiority of the proposed edge detection technique.

Response 1:

Some modifications have been made to the paper, and quantitative factors that can compare the accuracy of edge search are added. Please see section 4.1, 4.3 and 4.4 of the revision.

Superiority of the proposed edge detection technique: For one image in noisy environment, especially one edge-blur image in in noisy environment, when other edge detection operators or other wavelet filters are powerless, by suing of the length 8-4 rational coefficients biorthogonal wavelet filters and the algorithm proposed in this paper, edge information could be detected clearly and accurately. These can be proved by the results of multiple groups of comparative experiments shown as in section 4.1, 4.3 and 4.4 of the reversion, and the added quantify results shown as in Table 1, Table 2, Figure 13~Figure 15, Figure 24, Figure 34-Figure 37 of the revision.

Point 2: It is necessary to describe the specific application fields of the proposed edge detection technique.

Response 2:

The technique proposed in this paper is suitable for image edge detection in noisy environment. It is different from some other image denoising algorithms, which mainly pay attention to the weakening or removal of image noise, that is, the restoration of clear images from noisy background, and further more do some edge extraction or other operations based on the restoration clean images if necessary. The technique proposed in this paper is aiming at the noise and image edge signals themselves directly. It does not take into account the low-frequency information of the image overview, and directly use the correlation relationship between different scale coefficients reflected in the wavelet tree and Lipschitz exponential property of noise, to make the selection and judgment of noise and edge information in the multi-scale high-frequency region of the wavelet transform, so as to extract the edge information in the noisy environment.

The technique proposed in this paper is suitable for some industrial field environment (noisy environment), for the applications of extraction of target image edges for shape extraction and target positioning, etc. Especially, for the image with fuzzy edges caused by target high-speed motion in the field environment (noise environment), the technique proposed in this paper is still able to extract edge information and apply it to the applications such as target positioning.

Those descriptions have been added to the revision; please see the section of Abstract and Introduction (the first and second paragraphs also give the description for application field of Video-metrics) in the revision.

Point 3: It is difficult to understand the relationship between equation 2-2 and figure 4 because of figure 5. So, equation 2-2 and figure 4 should be explained in one place and then figure 5 should be explained behind equation 2-2 and figure 4.

Response 3:

The previous structure of the paper did interfere with the understanding of the relationship between equation 2-2 and figure 4. Therefore, some adjustments have been made to the structural arrangement to put equation 2-2 and figure 4 be explained in one place.

For specific adjustments, please see section 2.3.1 of the revision.

Reviewer 2 Report

This paper presented an edge detection algorithm with noise toleration using the correlation relationship between layers of wavelet transform coefficients. The authors contribution consists in

- remaining the wavelet transform coefficients by using activation function and

- eliminating other coefficients by performing outlier search judgment algorithm.

As such, the manuscript seems to me a sort of intuitive, minor adjustment over previous wavelet transform.

First of all, it is not clear what "fuzzy" in the article means. It appears only four times in the paper. In line 80 the authors refer to fuzzy image. I believe that the proposed solution actually targets a much more specific situation where images’ edges are blurred under some poor imaging condition. This should be clearly presented with an unambiguous term that is introduced since the abstract and the intro.

The authors consider only few algorithms based on wavelet coefficients. I think that a journal publication deserves much deeper literature analysis: a brief survey on all the noise removal approaches in the literature should be presented, including also the far more advanced algorithms as follows:

[r1] Highly Accurate Image Reconstruction for Multimodal Noise Suppression Using Semisupervised Learning on Big Data, in IEEE Transactions on Multimedia, 2018.

[r2] Beyond a Gaussian Denoiser: Residual Learning of Deep CNN for Image Denoising, in IEEE Transactions on Image Processing, vol. 26, no. 7, pp. 3142-3155, July 2017.

[r3] Learning Deep CNN Denoiser Prior for Image Restoration. CVPR, 2017.

On top of this, the authors should provide convincing arguments about the importance of methods based on wavelet transform estimation and why it is worth improving them, to emphasize the importance of the proposed solution and to prevent the reader to consider this as a pure academic exercise.

The authors comment about the aim of edge detection: "The aim of edge detection is to accurately extract edge information with a better noise suppression effect." However, this is not numerically and visually demonstrated in the paper.

I am also concerned about the experiments. The compared methods are outdated. It would be much better to see if the edges are able to be detected after removing noise. Thus, the proposed method should be compared with the far more advanced algorithms [r1]-[r3] to see the improvement of the proposed solution.

In summary, I think the changes above should be considered seriously.

Author Response

Point 1: First of all, it is not clear what "fuzzy" in the article means. It appears only four times in the paper. In line 80 the authors refer to fuzzy image. I believe that the proposed solution actually targets a much more specific situation where images’ edges are blurred under some poor imaging condition. This should be clearly presented with an unambiguous term that is introduced since the abstract and the intro.

Response 1:

It really does not explain clearly what the “fuzzy” means in the previous description of the paper. In this paper, “fuzzy” mainly refers to “motion blur”, which usually caused by the moving speed of the photographed target is too fast while the exposure time of the camera is not too short (the exposure frame frequency of the camera is less than the moving speed of the target). If improving the camera’s performance or selecting a camera with a fast enough frame frequency, it could really indeed reduce the problem of motion blur to some extent and improve the image quality. However, in many industrial applications, the selection of imaging equipment should take into account a variety of factors, as well as economic factors. What’s more, in the industrial field environment, the image quality is also easily affected by light, noise and other environmental factors. This paper is to explore how to extract the edge information from a moving fuzzy image with poor imaging quality, so as to realize target positioning and geometric shape analysis problems.

The image motion blur caused by the mismatch between camera exposure time and target motion speed can indeed be analyzed from the perspective of mathematical modeling, so as to restore and reconstruct the clear image. However, the goal of this paper is not for the restoration and reconstruction of the image, it is mainly for the extraction of edge information from noisy moving fuzzy image directly. Therefore, this paper did not give too much explanation to the mathematical model of motion fuzzy image, but focused more on the techniques of judgment for noise and image edge signal, and proposed one solving method for edge detection for noisy moving fuzzy image.

According to the comments on the review of the paper, the meaning of “fuzzy” in this paper, as well as the main research and application fields of this paper, have been revised and explained, please see the section of Abstract and Introduction in the revision.

Point 2: The authors consider only few algorithms based on wavelet coefficients. I think that a journal publication deserves much deeper literature analysis: a brief survey on all the noise removal approaches in the literature should be presented, including also the far more advanced algorithms as follows:

[r1] Highly Accurate Image Reconstruction for Multimodal Noise Suppression Using Semisupervised Learning on Big Data, in IEEE Transactions on Multimedia, 2018.

[r2] Beyond a Gaussian Denoiser: Residual Learning of Deep CNN for Image Denoising, in IEEE Transactions on Image Processing, vol. 26, no. 7, pp. 3142-3155, July 2017.

[r3] Learning Deep CNN Denoiser Prior for Image Restoration. CVPR, 2017.

On top of this, the authors should provide convincing arguments about the importance of methods based on wavelet transform estimation and why it is worth improving them, to emphasize the importance of the proposed solution and to prevent the reader to consider this as a pure academic exercise.

Response 2:

It really does not include enough deeper literature analysis in the previous description of the paper, and some supplements have been made to the revision.

We have read carefully the three papers and related articles given in the comments. The latest techniques including deep CNN structure, discriminative learning method, and semisupervised learning on big data, have indeed achieved very excellent image denoising effects, and those methods are worth learning by us to improve our future work.

In comparison, the differences and main research features of our proposed paper are as follows:

    The goal of the paper is not to recover a clean image from a noisy observation, but to make a trade-off judgment for noise signal and image edge information directly according to the characteristics of wavelet transform coefficients, to realize the extraction of edge information from a noisy image directly.

    For the image edge detection from noisy background, there are two main reasons for the selection of wavelet transform method. First is the correlation relationship between different scale coefficients reflected in the wavelet tree, and second is the Lipschitz exponential property of noise. Based on the two criteria, the paper proposed methods to realize separation of edge and noise signals in high-frequency region of wavelet transform.

Unlike the discrete Fourier transform (DFT) and discrete cosine transform (DCT), wavelet transform is not a fixed kernel transform. The selection of different wavelet bases will cause the different effects. In the research of the paper, it is found that with the length 8-4 rational coefficients biorthogonal wavelet filters proposed by the author and the algorithm proposed in this paper, excellent edge detection effect for noisy image can be achieved. It can be seen from the comparison experiments in the paper.

In view of these contents, the author makes some further supplements, revision and explanations. Please see the revision of this paper.

Point 3: The authors comment about the aim of edge detection: "The aim of edge detection is to accurately extract edge information with a better noise suppression effect." However, this is not numerically and visually demonstrated in the paper.

Response 3:

In the revision of this paper, the presentation methods to reflect the experimental results have been adjusted. Some quantitative data of detected edge position are added to reflect the accuracy of edge detection results.

Please see the section 4.1, 4.3 and 4.4 of the reversion, and the added quantify results shown as in Table 1, Table 2, Figure 13~Figure 15, Figure 24, Figure 34-Figure 37 of the revision.

Point 4: I am also concerned about the experiments. The compared methods are outdated. It would be much better to see if the edges are able to be detected after removing noise. Thus, the proposed method should be compared with the far more advanced algorithms [r1]-[r3] to see the improvement of the proposed solution.

Response 4:

Because this paper is not exploring the work for separating clean image from noisy background, it is hard to discuss the merits or disadvantages for image denoising effects. In order to illustrate the accuracy of edge extraction results, some quantitative data of edge location are added in the revision. Please see the section 4.1, 4.3 and 4.4 of the reversion, and the added quantify results shown as in Table 1, Table 2, Figure 13~Figure 15, Figure 24, Figure 34-Figure 37 of the revision.

Reviewer 3 Report

Authors present edge detection method for high-speed moving targets with different blurring effect by implementing proposed method which integrated the length 8-4 rational coefficients biorthogonal wavelet filter and the edge detection algorithm.  Authors’ work are appreciated and the proposed method could be useful based on the results demonstrated in this draft.  However, there are still some points which authors need to be clarified; 1. For Line 52-53, it said that image noise and image edge are both high-frequency information are both “high-frequency information”; however, for a shot image, the noise should be spatial high frequency with respect to objects in an image, and of course the edge would be, but it nothing relates to the “high-speed” moving.  Meanwhile, for the following discussion, no spatial frequency and/ or temporal frequency are discussed.  Authors should verify what “high-frequency information” means for here. 2. The definition of edge, mathematically in Line 56, said by authors is “the image edge presents local singularity “; however, for any optical imaging system, the aperture of the imaging system would always introduce diffraction effect and causing edge to be natural blurred.  Authors might consider to emphasis the proposed method is based on assumption for simplifying the mathematical model. 3. For this draft, the conditions given by authors regarding that the Noise and edge signal have a different Lipschitz exponent properties (in Line 139-141), this is essential point, are there any supporting document, reference and/or proof to support this point? 4. In Figure 2, L and H combinations are not defined.   5. Line 240-243, what does lower-index s mean for? 6. Section 3-1, Authors is suggested to improve the description of this section!  Parameter l is not defined and the connection with equations is missed. 7. Section 3-2 and 3-3, what dose function Q stand for?  What are the properties of Q?   8. Line 114 and Equation (3-3), there are typos 9. Line 169, “but the importance is getting stronger” what does it mean?

Author Response

Point 1: For Line 52-53, it said that image noise and image edge are both “high-frequency information”; however, for a shot image, the noise should be spatial high frequency with respect to objects in an image, and of course the edge would be, but it nothing relates to the “high-speed” moving.  Meanwhile, for the following discussion, no spatial frequency and/ or temporal frequency are discussed.  Authors should verify what “high-frequency information” means for here.

Response 1:

In this paper, “high-frequency information” is only means for the spatial frequency information; that is, for one digital image, edge information and noise information both belong to the mutation region of pixel gray value in spatial distribution. This paper only discusses edge detection techniques for one single frame image, so it has no relation with time frequency. And in this paper, “high-speed” means for the motion speed of the tested target. If the motion speed of the tested target is too fast while the exposure time of the camera is not too short (the exposure frame frequency of the camera is less than the moving speed of the target), it will cause a fuzzy edge effect on the digital image. This paper is to explore how to extract the edge information from a moving fuzzy image with poor imaging quality (noisy background), so as to realize target positioning and geometric shape analysis problems.

It really does not explain clearly in the previous description of the paper, and some supplementary explanations have been added to the revision. Please see the section of Abstract and Introduction

Point 2: The definition of edge, mathematically in Line 56, said by authors is “the image edge presents local singularity “; however, for any optical imaging system, the aperture of the imaging system would always introduce diffraction effect and causing edge to be natural blurred.  Authors might consider to emphasis the proposed method is based on assumption for simplifying the mathematical model.

Response 2:

Just as the reviewer mentioned, for any optical imaging system, the aperture of the imaging system would always introduce diffraction effect and causing edge to be natural blurred, and the method proposed in this paper is based on assumption for a simplifying mathematical model. That is, for one clearly digital image, gray value of one image edge pixel will present in the form of “step function” relative to the other adjacent background pixels. While in some cases, for one blur digital image, gray value of one image edge pixel will be approximated in the form of “slope line function” relative to the other adjacent background pixels.

For the mentions before, some supplementary explanations have been added to the revision. Please see the third paragraph of the section of Introduction in the revision.

Point 3: For this draft, the conditions given by authors regarding that the Noise and edge signal have a different Lipschitz exponent properties (in Line 139-141), this is essential point, are there any supporting document, reference and/or proof to support this point?

Response 3:

It is really an essential point that the noise and edge signal have a different Lipschitz exponent properties, and this is also an important basis for the edge detection algorithm proposed in this paper. There are certain supporting document and reference to support this point, and in the revision, some supporting documents have been added. Please see the revision paper.

Point 4: In Figure 2, L and H combinations are not defined.

Response 4:

In the figure 2, L means for “Low-pass”, H means for “High-pass”; LH, HL, LL and HH respectively means for “horizontal direction Low-pass and vertical direction High-pass”, “horizontal direction High-pass and vertical direction Low-pass”, “horizontal direction Low-pass and vertical direction Low-pass” and “horizontal direction High-pass and vertical direction High-pass” field of wavelet transform. The lower-index denotes different scales of wavelet transform.

These explanations have been added to the revised version of the paper. Please see the figure 2 in the revision.

Point 5: Line 240-243, what does lower-index s mean for?

Response 5:

The lower-index s denotes a certain scale of wavelet transform. This explanation has been added to the revised version of the paper. Please see line 299-301 in the revision.

Point 6: Section 3-1, Authors is suggested to improve the description of this section!  Parameter l is not defined and the connection with equations is missed.

Response 6:

In equation (3-1), parameter l means for the time delay term. For one specified wavelet filter design, we proposed setting the time delay term l=0, in order to get the filters’ relationship such as shown in equation (3-1) and (3-4) in the revision paper.

A more clear explanation has been made and please see section 3.1 in the revision paper.

Point 7: Section 3-2 and 3-3, what dose function Q stand for?  What are the properties of Q?  

Response 7:

Here, function Q is the higher order equation for the variable T, that is, function Q(T) can be represented as “1+a1*T+a2*T^2+a3*T^3+…+am*T^m”, and index m is determined by the wavelet filter vanishing moment. This design method is based on the theory of Daubechies orthogonal compactly-supported wavelet filter, which can be referred to in detail. As for the details of filer design method, more have been presented in the previous published papers by the author and could be referred to for details.

Some explanations have been added to the revision paper, and please see the section 3.2 of the revision.

Point 8: Line 114 and Equation (3-3), there are typos

Response 8:

The typos in line 114 has been modified, please see line 160-163 in the revision paper. But we examined Equation (3-3) (previous paper) carefully and didn’t find some typos. In Equation (3-1), 2z^-l, the power ” l” is the small letter of L, means for the time delay term, and in Equation (3-1), it can take any number. In Equation (3-3), z^-1*H(-z), the power is the Arabic numeral “-1”, which is a specific design for the filter.

Point 9: Line 169, “but the importance is getting stronger” what does it mean?

Response 9:

The previous statement is really not appropriate. This means that those wavelet transform coefficients contain more information, so they have more significance and their importance is greater.

The presentation has been modified. Please see section 2.3 in the revision paper.

Round 2

Reviewer 1 Report

Manuscript ID: sensors-401640

Title: Algorithm Design for Edge Detection of High-speed Moving Target Image under Noisy Environment

Review Comments: Minor revision

This paper deals with edge detection of high-speed moving target image. It appears to be a well-structured paper, with minor modifications as follows.

You put equation 3-3 within local language. So that it is better to modify by international language.

There are too many references on paragraph from lines 340 to 344. I think you should just mark the reference explicitly cited.

Author Response

Point 1: You put equation 3-3 within local language. So that it is better to modify by international language.

Response 1:

The description of equation 3-3 has been modified in the revision. Please see line 318-319 of the revision. Thanks very much for suggestion. 

Point 2: There are too many references on paragraph from lines 340 to 344. I think you should just mark the reference explicitly cited.

Response 2:

The citation of references has been adjusted to make a more explicit reference relationship, and 3 less-important references have been removed. Please see line 340-345 of the revision. Thanks very much for suggestion. 

Reviewer 2 Report

The authors have taken most of my comments. Now the quality of the paper has been improved and my concerns have been cleared.

Author Response

Reviewer 2:

Point 1: The authors have taken most of my comments. Now the quality of the paper has been improved and my concerns have been cleared.

Response 1:

Thank you very much for your advices and approval, and we will try our best in the future.